# Proliferative arrest induces neuronal differentiation and innate immune responses in normal and Creutzfeldt-Jakob Disease agent (CJ) infected rat septal neurons

Nathan Pagano[1], Gerard Aguilar Perez[1¤], Rolando Garcia-Milian [2], Laura Manuelidis [1*]

1 Section of Neuropathology Surgery, Yale University Medical School, New Haven, Connecticut, United States of America, 2 Bioinformatics Support Hub, Yale Medical Library, Yale School of Medicine, New Haven, Connecticut, United States of America,

¤Current address: Laboratory of Molecular Neuroscience, German Center for Neurodegenerative Diseases (DZNE), Berlin, Germany
* laura.manuelidis@yale.edu

## Abstract

Rat post-mitotic septal neurons, engineered to reversibly proliferate and arrest under physiological conditions, can be maintained for weeks without cytotoxic effects. Nine representative independent cDNA libraries were made to evaluate global arrest-induced neural differentiation and innate immune responses, e.g., upregulated interferon (β-IFN) RNA, that were previously identified in normal uninfected and Creutzfeldt-Jakob Disease agent (CJ) infected septal neurons. This reversible cell model encompassed a non-productive latent (CJ−) and a highly infectious (CJ+, 10 logs/gm) state. Arrest of normal uninfected neurons upregulated a plethora of anti-proliferative transcripts and known neuronal differentiation transcripts (e.g., Neuregulin-1, GDF6 and Prnp). As expected, many activated IFN innate immune genes were simultaneously upregulated (e.g., OAS1, ISG20, CD80, cytokines, chemokines and complement) along with clusterin (CLU) that binds misfolded proteins. Arrest of latently infected CJ− cells induced even more profound global transcript differences. CJ+ cells markedly downregulated the anti-proliferative controls seen in arrested normal cells. CJ+ infection also suppressed neuronal differentiation transcripts, including Prnp which is essential for CJ infection. In contrast, IFN and cytokine/chemokine pathways were strongly upregulated. Analysis of the 342 CJ+ unique transcripts revealed additional innate immune and anti-viral-linked transcripts, e.g., Il17, ISG15, and RSAD2 (viperin). These data show: 1) innate immune transcripts are produced by normal neurons during differentiation; 2) CJ infection enhances and expands anti-viral responses; 3) non-productive latent infection can epigenetically imprint many proliferative pathways to thwart complete arrest. This rare cell model of latent infection is fundamental for interrogating triggers of late onset disease that are also relevant for Alzheimer's Disease. Peripheral human blood and

**Data availability statement:** All relevant data are within the manuscript and its Supporting Information files.

**Funding:** This work was supported by the Hanna Howard Fund and a gift from the William Prusoff Foundation to LM. Access to Partek Flow (Illumina), Qlucore Omics Explorer (Qlucore), and Ingenuity Pathway Analysis (Qiagen) was provided and sponsored by the Cushing/Whitney Medical Library, Yale School of Medicine.

**Competing interests:** The authors have declared that no competing interests exist.

intestinal myeloid cells that are latently infected may also be conditionally stimulated in vitro to produce CJ+ linked diagnostic transcripts.

## Introduction

There are relatively few papers analyzing cDNA libraries of normal differentiating neurons in culture as they transition from a proliferative to an arrested G1 state. Post-mitotic neuron differentiation in animals takes place over many days with differentiation times determined by species and neuronal cell type [1]. Studies of neuronal differentiation in brain are complicated by cellular and humoral complexity. Neuro-ectodermal cells (as astrocytes), along with vascular endothelium, microglia, and blood borne factors (serum molecules and hormones) can have multiple effects on core neuronal differentiation and cell fate programs. Cultured rat post-mitotic septal (SEP) neurons immortalized with a temperature sensitive (ts) SV40 construct provide a powerful model for studies of neural differentiation over a 3–4 week period without these complexities [2,3]. We are not aware of any reports of post-mitotic neurons where differentiation can be reversibly turned on and off for weeks, a feature that can be particularly informative for understanding latent viral infections.

SEP cells proliferate at 33°C and can be physiologically arrested at 37.5°C in 2% serum. Arrested cells stop synthesizing DNA by 4–6 days post-arrest, as assessed by BrdU incorporation in nuclei, and these cells can be maintained in an arrested state by refeeding every 2 days for >75 days without reverting to a proliferative state [4]. During this time, cell-to-cell contacts and neural markers increased along with host prion protein (PrP) assayed on western blots and host gene transcripts (Prnp) evaluated by RT/qPCR [4,5]. PrP is a neural differentiation marker that increases during synaptogenesis and localizes to membranes by electron microscopy [6]. PrP is also a host susceptibility factor required for infection in Creutzfeldt-Jakob Disease (CJD) and other Transmissible Spongiform Encephalopathies (TSEs) [7], and its misfolded or amyloid form is a marker of brain disease. Previous RT/qPCR and Western blot protein studies of normal SEP cells documented virtually complete cell arrest by cessation of nuclear DNA synthesis using labeled nucleotides, in addition to concomitant neurodifferentiation. For example, uninfected neurons exhibited a 20–28-fold upregulation in Prnp expression by RT/qPCR with a sustained 8-fold increase in PrP on Western blots [4,5]. To reveal a broader span of commonly linked molecular transcripts, large scale cDNA libraries were made and they revealed abundant global networked pathways induced during normal neuronal arrest.

These previous RT/qPCR studies of normal SEP neurons also showed that arrest induced an 8-fold upregulation of β-interferon (β-IFN) RNA, an unexpected classical innate immune response [4] not previously shown to be produced by normal neurons. The cDNA libraries were interrogated to confirm and expand additional immune changes in normal neurons in these samples during neuronal differentiation.

Therefore, DNA libraries here were tested to substantiate functional IFN activated immune transcripts produced by normal neurons during differentiation.

Parallel studies of normal SEP neurons infected by rat passaged human FU-CJD agent (CJ) [5,8] provided a valuable comparison. During arrest CJ infected SEP cells produced persistent high titers of the infectious agent (9.7 logs/gm for 120 days) along with pathologic PrP amyloid that failed to produce visible cytotoxic or neurodegenerative stigmata observed in high infectivity CJD brain. In brain, but not infected myeloid cells, misfolded PrP amyloid appears late, 80 days after the CJ agent has silently replicated to high levels, e.g., 8.5 logs/gm brain [9]. This is inevitably lethal. In contrast, highly infectious (CJ+) SEP cells with no pathology, allowed to proliferate at 33°C, rapidly lost 5 logs of infectivity, i.e., <1 infectious dose/gm [5]. To determine if these proliferating SEP cells with no detectable infectivity (CJ−) were latently infected, they were re-arrested. Again, they produced very high levels of infectious agent (10 logs/gm) without pathological stigmata, proving that latent, silent non-productive infection remained in the CJ− proliferating cells. Re-arrested CJ+ cells rapidly displayed PrP amyloid along with 40–60-fold elevated β−IFN transcripts versus 6–8-fold in uninfected cells [4]. Multiple RNA samples from these previously characterized proliferating CJ− and re-arrested CJ+ cells were used to make libraries for bioinformatic analysis of additional transcripts and pathways that can be altered during non-productive latent, and productive CJ+ infection.

It is important to appreciate the importance and medical and veterinary relevance of the long latent phase in TSEs. Latent infections can be notoriously extended in people and cannibalism associated kuru agent infections in New Guinea can take over 10 years to produce neurological disease [10]. Sporadic CJD (sCJD) contaminated growth hormone injected peripherally can also remain latent for at least 30 years [4], with recorded latencies of 38 years [11]. Moreover, unlike kuru that disappeared with the cessation of ritual cannibalism, a subset of the population exposed to sCJD contaminated growth hormone, in addition to people eating meat infected with the bovine TSE agent, may continue to harbor silent, but potentially lethal infections. Latent states of infection in TSEs (silent, non-productive states) have largely been forgotten in studies focusing on late-stage PrP amyloid conversion.

Latency in TSE infections were substantiated by experimental animal studies. Endemic sheep scrapie, a neurodegenerative disease recognized for over 500 years, human sporadic Creutzfeldt-Jakob Disease (sCJD), and epidemic "mad cow" disease (BSE) belong to the group of TSEs [12]. Fundamental infectious and pathologic features of TSEs were first demonstrated in 1936 by serial passage of sheep scrapie brain [12]. In these experiments the infectious agent was referred to as a virus or a slow virus because of its undeniably virus-like biology (see below). Because the molecular nature and strain-defining attributes of the infectious particle remain unresolved, we use the term infectious agent. Notably, misfolded non-infectious PrP amyloid can't be distinguished from infectious PrP amyloid, e.g., [13], and TSE infectious particles, as viruses, require nucleic acids [14]. The word virus below is used according to the authors' designation.

Wilson showed the scrapie virus (in clarified brain homogenates) efficiently passed through 0.41μ filters, and caused disease via various inoculation routes, e.g., intracerebral, intravenous, cutaneous [15]. Successful transmission of the sheep scrapie virus to rodents in the 1960s, critical for pathogenesis studies, showed the incubation time to disease after inoculation (latency) was dependent on dose, route and species susceptibility, with lymphoid tissue infections (a latent reservoir) preceding brain infections by 10 weeks [16]. Transmission of sCJD to small animals in the 1970s added other medically relevant findings such as infection by white blood cells and lack of maternal transmission [17]. As in scrapie, different CJD agents were identified by their latency and the distribution and severity of brain lesions [12,18]. In normal mice the sCJD agent is slow and produces minimal lesions restricted to the medial thalamus while the Asiatic FU-CJD strain replicates rapidly and produces widespread destructive lesions. Despite this difference in virulence, with a 10,000-fold difference in infectivity titers, both human CJD agents show indistinguishable PrP amyloid banding patterns with only a 10-fold difference in amount. Indeed, PrP amyloid Western blot band patterns vary with cell type and this fails to modify the agent strain biology [18,19]. The pathogenetic features and importance of latency in sCJD and FU-CJD were further demonstrated using a classical viral vaccination strategy. The inoculated low infectivity slow sCJD agent prevented

superinfection by the high infectivity fast FU-CJD agent in mice, and this was reproducible in cell cultures where various scrapie agents were also compared [20–22]. These strain findings are of relevance because in principle, latent infection could underlie resistance to infection by more virulent infectious.

Since the differentiation of various types of neurons is exquisitely connected to specific sequential times of development, we compared representative early and later post-arrest days in parallel studies of uninfected and CJD infected SEP cells. To ensure the high reproducibility of major transcript and pathway changes, different independent SEP cell passage groups at multiple days after arrest were compared using the most relevant and representative time points based on previous RT/qPCR, protein and shared cellular marker studies [3,4]. Prior detailed biological and molecular arrest studies including DNA synthesis, RT/qPCR and Western blots on these and other independent SEP cell experiments have been detailed previously [3–5].

## Materials and methods

### Cell culture characterizations

All experiments and cultures used to make cDNA libraries were previously described in detail [4]. Briefly, low passage post-mitotic rat Septal (SEP) neurons, immortalized with a temperature sensitive ts-SV40 T antigen (subclone e422, [1]), were passaged every 4 days at 1:4 in a proliferating state at 33°C in 10% serum-DMEM. To induce and maintain arrest, cells were cultured in 2% serum-DMEM at 37.5°C and refed every 2 days without splitting. Normal arrested cells became 95% stationary by day 2, shown by cessation of BrdU incorporation into DNA and loss of SV40 T antigen protein (Tag) e.g., [2]. With arrest, prion protein (PrP) progressively increases to 15–20x the normal uninfected level in SEP cells from passages 7, 17 and 25 [4], and these arrested and parallel controls were used here for cDNA libraries and analyses. Briefly, thawed untreated normal cells from these different passages were either allowed to proliferate, or maintained in an arrested state for the indicated days of each passage set. Independent groups of parallel proliferating and arrested cells from the same day were compared. Proliferating and arrested cells from these 3 independent passages showed comparable patterns of PrP, SV40 Tag and β-IFN changes. Fig 1A (see Results) summarizes these proliferating control uninfected cells in light green (Prol/NI) with their parallel arrested cells sampled on the same day in darker green (Arst/NI).

Fig 1A also shows two CJD groups with control proliferating cells (Prol/CJ−) in light orange, and their parallel re-arrested counterparts in dark orange (Arst/CJ+). More specifically, proliferating SEP cells that had been infected with FU-CJD rat brain homogenate [8] that were arrested for 75 days attained a high titer of 5 tissue culture infectious doses per cell (TCID), equivalent to 5E9 infectious particles/gm where 1gm of brain contains $10^9$ cells in standardized assays as described [5,23,24]. These high infectivity cells were frozen at passage 17 (p17). They were then thawed and allowed to proliferate for 3 additional passages. Even with 1 passage of release from arresting conditions, the high CJD infectivity of 5 TCID/cell was substantially lost and reduced to 2 TCID/1,000 cells, i.e., a loss of 3 logs [5]. Thus, proliferating SEP cells are designated Prol/CJ− to indicate they lost all detectable infectivity after further passaging these samples [4]. These CJ− controls were pertinent because the effects of previous high FU-CJD infection might alter standard normal SEP cell characteristics. In the current experiments, at passage 3 after thawing, proliferating FU-CJD SEP cells were re-arrested and maintained in parallel with their CJ− controls. By 6 days and 34 days post-arrest their infectivity increased to 2 and >10 TCID per cell respectively (>10 logs/gm) in standard GT1 indicator cell assays as depicted [4]. TCID infectivity assays correlate well with $LD_{50}$ animal agent titrations, and re-inoculation of infected cells also reproduce their strain-specific characteristics in mice. These high infectivity FU-CJD SEP cells are designated Arst/CJ+. Selection of passage and sample days for both normal and CJD groups here were based on protein and RT/qPCR cDNA analyses as graphically detailed [4] for elevated β-interferon, Prnp and absent Tag RNAs in addition to relevant protein changes including protease resistant PrP amyloid.

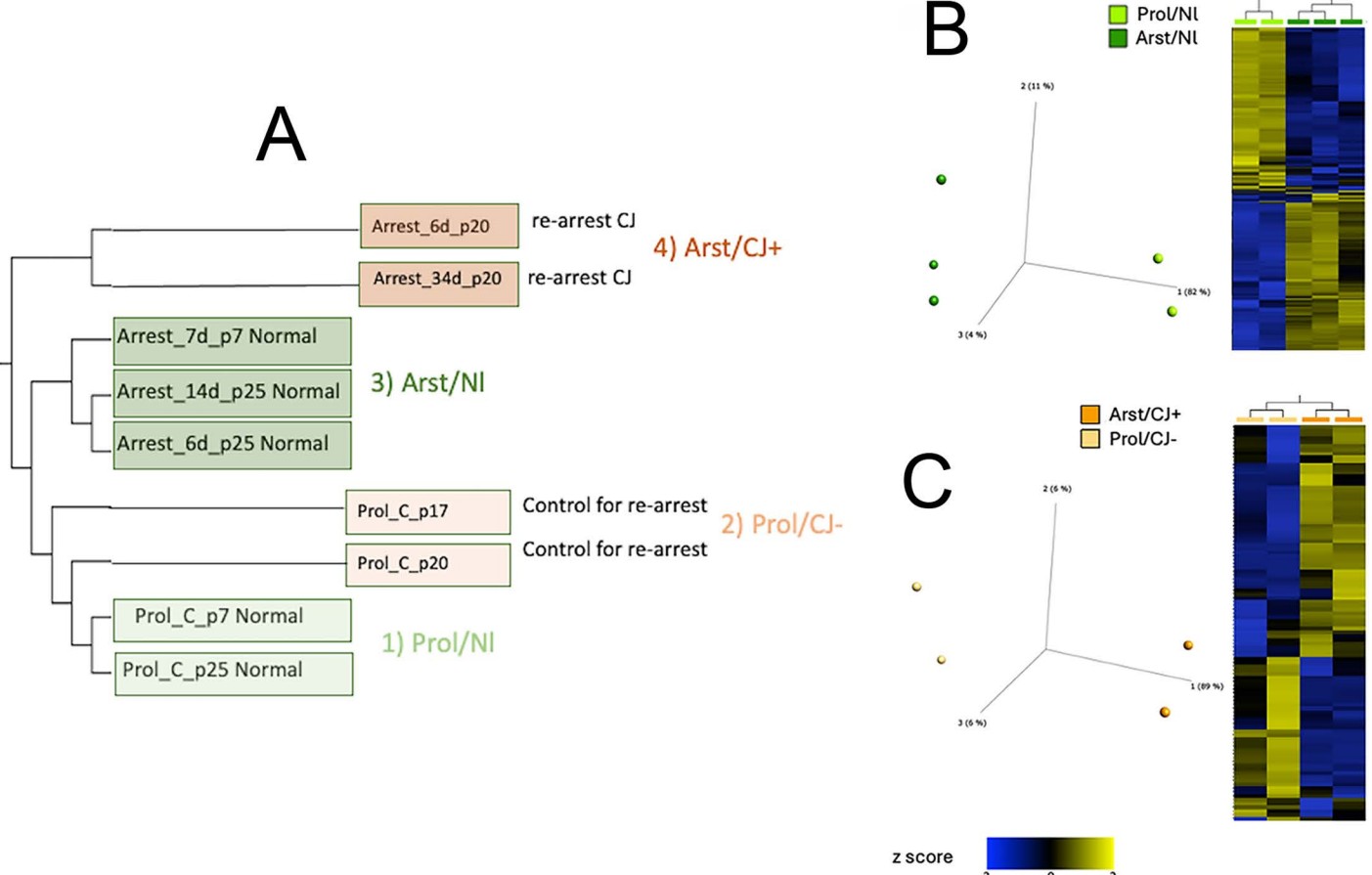

**Fig 1. A): Hierarchical clustering of different passages (p) and days based on 22,251 normalized genes.** Proliferating NI controls (Prol/NI) light green, p7 & p25) were closely related to Proliferating CJ− neurons (Prol/CJ−) light red (p17 & p20). Arrested NI cells (Arst/NI) darker green and arrested CJ (Arst/CJ+) darker red were also more closely related to each other (groups 3 & 4 respectively) than to their proliferating counterparts. **B**): Principal component analysis (PCA) and hierarchical clustering heatmap comparing NI proliferating and NI arrested cell sets. DESeq2 differential analysis (FDR p < 0.05, Fold change > |2|) resulted in 379 differentially expressed genes between these 2 groups (205 downregulated and 174 upregulated). **C**) Principal component analysis (PCA) and hierarchical clustering heatmap comparing CJ− proliferating and CJ+ arrested cell sets yielding a total of 112 differential genes. DESeq2 differential analysis (FDR p < 0.05, Fold change Log2ratio > |0.5|) resulted in differentially expressed genes between these 2 groups (45 downregulated and 67 upregulated).

## RNA sequencing

Brett Robb and staff at New England Biolabs generously constructed the cDNA libraries and performed the RNA sequencing for the above samples. Between 72–92 ng of RNA per sample was subjected to rRNA depletion via the NEBNext® rRNA Depletion Kit v2 (Human/Mouse/Rat; NEB #E7400) using the manufacturer's protocol except that a Monarch® RNA Cleanup Kit (10 µg; NEB# T2030) was used to purify the rRNA-depleted RNA instead of magnetic beads. The rRNA-depleted RNA was converted into Illumina-compatible DNA libraries via the NEBNext® Ultra™ II Directional RNA Library Prep with Sample Purification Beads kit (NEB# E7765) using the manufacturer's protocol for use with rRNA-depleted RNA and NEBNext® Multiplex Oligos for Illumina® (NEB# E6448). Samples were pooled and sequenced 2x150 nt paired-end on an Illumina® Novaseq 6000 instrument.

## RNA bioinformatic analysis

PartekTM FlowTM software (version 11) bulk RNA-seq pipeline was used for data analysis. In summary, trimmed reads were aligned to the Rnor_6.0 (rn6) genome reference using STAR [25] (version 2.7.8), and subsequently Partek E/M algorithm [26] was used to count reads mapping to the genes from Ensembl release 95. We applied the DESeq2 normalization method and top 1000 genes by variance were analyzed for PCA. Heatmaps show row-normalized relative gene expression z-scores across columns. Qlucore Omics Explorer (Qlucore, Lund, Sweden), a dynamic, interactive visualization-guided bioinformatics program with a built-in statistical platform was used for data visualization. Differential expression was performed using the DESeq2 [27] method. The cutoff value to select differentially expressed genes was Log2ratio ≥ |0.5|, FDR $p < 0.05$ [28]. A copy of the dataset was stored in the National Center for Biotechnology Information, Gene Expression Omnibus database: GSE272571

## Functional analysis

Overrepresented pathways, biological functions, upstream regulators, and biological networks were identified using Ingenuity Pathway Analysis (QIAGEN Redwood City, CA) knowledgebase and g:Profiler [29]. For this, differentially expressed gene identifiers are mapped and analyzed using a one-tailed Fisher Exact Test, (FDR $p < 0.05$). In addition, Gene Set Enrichment Analysis (GSEA Mac App Broad Institute Inc.) [30] was performed using DESeq2-normalized counts with gene set permutation and otherwise default settings. By using the Rat_Gene_Symbol_Remapping_Human_Orthologs-MSigDB v7.2 chip, gene counts were tested against the human hallmark (H), canonical pathways (C2), gene ontology (C5), and immunologic signature (C7) gene sets (version 7.5.2). Gene sets with false discovery rates (FDR) $p < 0.25$ were considered enriched as suggested by GSEA developers. The overlap and connections between the resulting different gene sets were produced by the Enrichment Map Plugin (http://baderlab.org/Software/EnrichmentMap) for Cytoscape 3.8, considering a value of FDR $p < 0.05$. The nodes were joined if the overlap coefficient was ≥ 0.375.

## Results

### Hierarchical clustering of 9 independent cDNA libraries

For comparative analysis, 9 independent SEP cell cDNA libraries were made from the uninfected and infected cell RNA extracts in experiments using different SEP cell passages for arrest. These libraries were chosen based on the day of the initial increase in Prnp (by RT/qPCR and PrP amyloid western blots), and β-IFN transcripts. Both transcripts rose steeply by 6–8 days after arrest. Arrested cell RNAs and their parallel proliferating controls (Fig 1) were assayed sequentially every 4–6 days for up to 35 days post-arrest [4] at the selected passages and days indicated in Fig 1A. Representative libraries, colored green for uninfected, and red for CJ agent infected, fell into distinct proliferating (lighter shades) and arrested (darker shades) groups (Fig 1A). The proliferating uninfected normal (NI) control sets, sampled at 2 different passages (p7 and p25), fell into group 1, and are designated Prol/NI. These NI proliferating transcripts were closely related to CJ− proliferating transcripts in group 2 (p17 and p20). Remarkably, despite their prior infection, these CJ− proliferating cells revealed only 19 differentially expressed transcripts (8 up & 11 down regulated). In contrast, comparison of both arrested groups 3 and 4 (Arst/NI and Arst/CJ+) contained 379 differentially expressed genes. Fig 1B shows the divergence of the uninfected NI groups 1 and 3 by principal component analysis along with a clustering of reproducible Prol/NI vs Arst/NI differences in independent experimental samples (group 2 proliferating vs group 3 arrested). Notably, the Arst/NI vs Prol/NI sets contain more downregulated (blue) than upregulated (yellow) genes in each of the two p25 Arst/NI RNAs samples at different days in culture (6 and 14 days). An independent second repeat experiment from p7 also has a highly similar cDNA profile as shown in the Fig 1B. As in the Arst/NI controls, Arst/CJ+ cells again segregate in a distinct group from Prol/CJ− cells. However, as shown in Fig 1C the CJ differential analysis resulted in only 112 differences as compared to 379 in uninfected SEP cells. Moreover, whereas downregulated genes dominated in the Arst/NI cells, 60% of the transcripts in CJ+ arrested cells were upregulated.

## Comparison of proliferating uninfected and CJ− cDNAs

While there were only 19 differences between proliferating NI and CJ− cell sets, several of them were relevant and suggested stable transcriptional changes caused by prior infection. The Prol/CJ− transcripts reproducibly contained upregulated acute phase protein LPB (up 7.2-fold) involved in innate immunity, and inflammation associated complement C1s (up 3.5-fold in the Prol/CJ− set). Mt1m was also upregulated 5.2-fold, and this zinc binding protein is high in several brain regions and localized to the perinuclear region where PrP amyloid fibrils aggregate in infected cells [6]. Six downregulated genes were also notable, including ApoE (down 5.8 fold), Adamts8 metallopeptidase (down 5.8 fold), and Cavin4 (down 9.1 fold) that encodes a Golgi and plasma membrane caveolar protein in synaptic vesicles, and possibly linked to vacuolar "spongiform" dendritic and synaptic changes in TSEs [31]. Tspan7, a transmembrane glycoprotein functionally involved in many different viral infections [32] was downregulated 10 fold, and neuron development genes Tenm3 and Dpysl3 were down 19.4 fold and 26.8 fold respectively in Prol/CJ− vs Prol/NI cells. To best appreciate the diversity and extent of specific changes in the 4 major groups, differential changes between uninfected SEP cells (Arst/NI and Prol/NI) are analyzed first, and then the CJ infected samples compared.

## Arrest produces abundant transcriptional changes in uninfected SEP cells

To assess reproducibility of changes we compared the 25 most downregulated transcripts in each of the 3 arrested NI cDNA sets. All these genes were present in at least 2 of these 3 independent samples regardless of function (S1 Table). The most downregulated transcripts (5–53-fold lower) are suppressed during DNA replication and cell cycle progression, including cell cycle checkpoints, histone synthesis (with 4 different histones lower by 6.5 to 28-fold) and spindle microtubule formation. A total of 18/25 downregulated genes localized to the nucleus. The longest NI arrest sample (p25 at 14 days) showed all these genes were downregulated whereas the shorter 6 and 7 day arrested samples displayed some variants (6 and 8 respectively) including two histone transcript clusters that were absent as they normally would be during the S phase of growth. Other minor fold variations in the top 25 transcripts, such as Cenpf, are also consistent with an initial lack of synchrony of G1, S and the shorter mitotic cell cycle phases, and the increased time needed to achieve a complete and stable synchronous arrest in G1. The total differential RNA-seq data further emphasizes multiple processes involved in SEP neuronal arrest were orchestrated precisely and sequentially, as during transitions though DNA replication and mitosis, and all were educed by physiological arrest. Upregulated top genes in Arst/NI were uniformly changed in all three independent samples (Table 1). Three nuclear anti-proliferative upregulated transcripts are seen with diverse other transcripts localized to the plasma membrane or extracellular space, at least *in vivo* (C1s). Ingenuity pathway analysis showed a very complex network of many replication-related genes and further demonstrated the extent of anti-proliferative upregulated genes in the entire data set. The entire Arst/NI data can't be graphed with any visual clarity due to all the participating pathways, so an example (from passage 7, day 7 transcripts) is shown in Fig 2.

### Neural differentiation genes are upregulated with arrest

18 highly upregulated transcripts were consistently represented in all three Arst/NI samples at different times post-arrest (Table 1). These upregulated cDNAs were more diversified functionally than those that were downregulated. They included the 1,023-fold upregulated retinol binding protein 2 (Rbp2) active in neuronal differentiation, a calcium channel (Clec2d2) involved in neuronal networks, and Sushi (Svep1) a transcript which encodes a multidomain adhesion protein involved in epidermal differentiation [33] but not yet evaluated in neuronal differentiation. The entire non-selected set of genes highlighted additional neuronal characteristics. Fig 3 shows the 4 transcripts with the strongest neuronal effects (darkest red) and 11 other major transcripts activating differentiation of neurons listed (lighter red). Of these, neuregulin 1 (NRG1) and angiotensin receptor 2 (AGTR) promote neurogenesis but have other non-neural actions. However, upregulated neuregulin HES6* (Fig 3), a helix-loop-helix transcription repressor that promotes neuronal differentiation

**Table 1. Consistent upregulation of 18 genes (Log2Ratio) in 3 independent Arst/NI samples.**

| # | Rat symbol | Entrez Gene Name | P7, 7d | | P25, 6d | | P25, 14d | | Location |
|---|---|---|---|---|---|---|---|---|---|
| | | | FDR(p) | Log2Ratio | FDR(p) | Log2Ratio | FDR(p) | Log2Ratio | |
| 1 | Rbp2 | retinol binding protein 2 | 3,67E-02 | 7,855 | 9,75E-03 | 10,813 | 4,46E-03 | 11,341 | Cytoplasm |
| 2 | Sulf2 | sulfatase 2 | 1,61E-04 | 3,498 | 2,60E-11 | 5,515 | 5,20E-12 | 5,668 | Plasma Membrane |
| 3 | Clec2d2 | C-type lectin domain family 2, member D | 1,42E-05 | 5,549 | 2,54E-04 | 5,329 | 1,54E-04 | 5,445 | Plasma Membrane |
| 4 | Itm2a | integral membrane protein 2A | 2,04E-03 | 3,919 | 9,20E-06 | 5,03 | 1,07E-06 | 5,344 | Plasma Membrane |
| 5 | Cfh | complement factor H | 3,35E-02 | 2,701 | 2,61E-06 | 5,007 | 1,15E-07 | 5,506 | Extracellular Space |
| 6 | Mmp13 | matrix metallopeptidase 13 | 6,42E-16 | 6,419 | 1,70E-07 | 4,827 | 4,76E-08 | 5,017 | Extracellular Space |
| 7 | Sfrp4 | secreted frizzled related protein 4 | 1,17E-02 | 3,079 | 2,48E-04 | 4,092 | 6,43E-05 | 4,329 | Plasma Membrane |
| 8 | Ephx1 | epoxide hydrolase 1 | 2,04E-03 | 3,141 | 2,16E-05 | 3,969 | 1,14E-05 | 4,031 | Cytoplasm |
| 9 | Ccdc80 | coiled-coil domain containing 80 | 4,02E-02 | 2,089 | 1,70E-07 | 3,899 | 6,28E-07 | 3,737 | Nucleus |
| 10 | Cdkn1a | cyclin dependent kinase inhibitor 1A | 4,55E-03 | 2,597 | 3,82E-03 | 3,757 | 6,71E-03 | 3,611 | Nucleus |
| 11 | Svep1 | sushi, von Willebrand factor type A, EGF & pentrax. | 9,95E-05 | 3,588 | 6,45E-05 | 3,618 | 9,41E-07 | 4,196 | Cytoplasm |
| 12 | Clu | clusterin | 1,12E-06 | 4,097 | 7,05E-07 | 3,585 | 1,86E-06 | 4,028 | Cytoplasm |
| 13 | Abca1 | ATP binding cassette subfamily A member 1 | 4,92E-04 | 3,37 | 2,93E-04 | 3,477 | 1,54E-04 | 3,579 | Plasma Membrane |
| 14 | Mmp2 | matrix metallopeptidase 2 | 1,19E-04 | 3,108 | 1,32E-05 | 3,419 | 6,40E-06 | 3,473 | Extracellular Space |
| 15 | Plk2 | polo like kinase 2 | 3,35E-02 | 2,759 | 8,51E-03 | 3,177 | 1,18E-02 | 3,05 | Nucleus |
| 16 | Tnn | tenascin N | 3,83E-04 | 2,852 | 1,70E-03 | 2,964 | 2,53E-03 | 2,867 | Plasma Membrane |
| 17 | C1s | complement C1s | 3,76E-04 | 3,22 | 4,52E-03 | 2,891 | 5,72E-03 | 2,845 | Extracellular Space |
| 18 | Edil3 | EGF like repeats and discoidin domains 3 | 3,29E-02 | 2,492 | 1,19E-02 | 2,78 | 1,88E-03 | 3,127 | Extracellular Space |

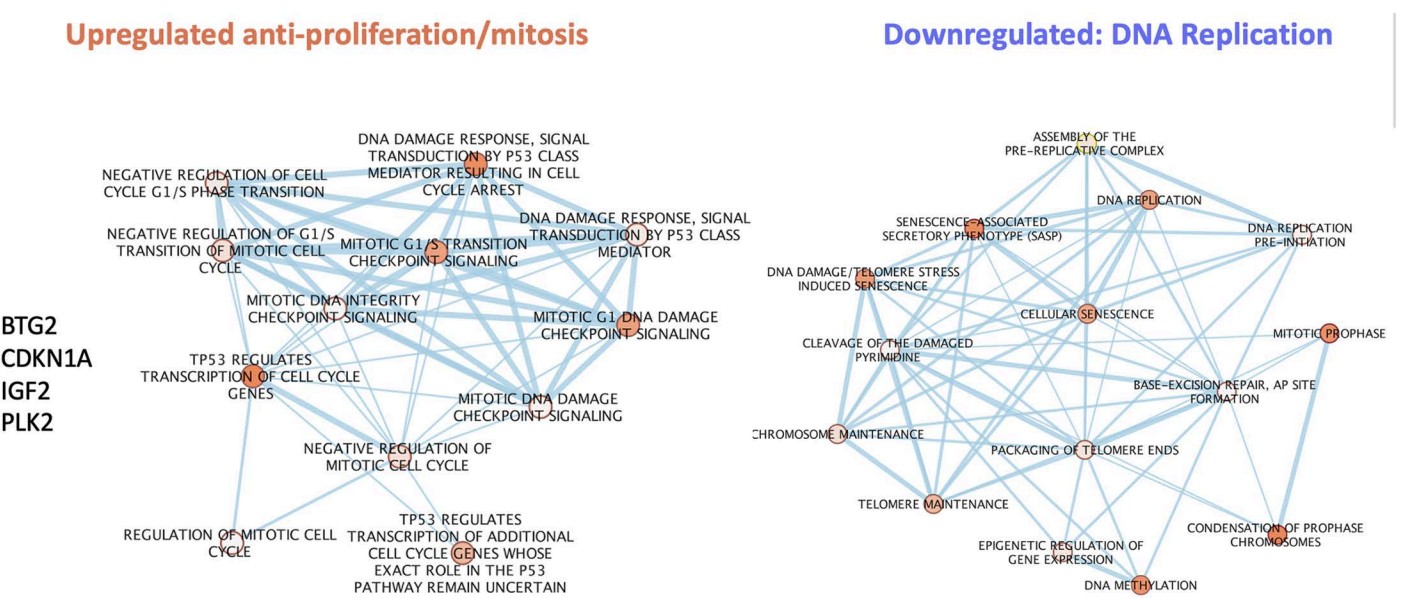

**Fig 2. Overrepresentation and clustering analysis of enriched pathways contributing to arrest with upregulated and downregulated genes of the p7 day 7 normal sample.** g:Profiler web server was used for overrepresentation analysis. Enriched signatures were analyzed with Enrichment map application and visualized on Cytoscape.

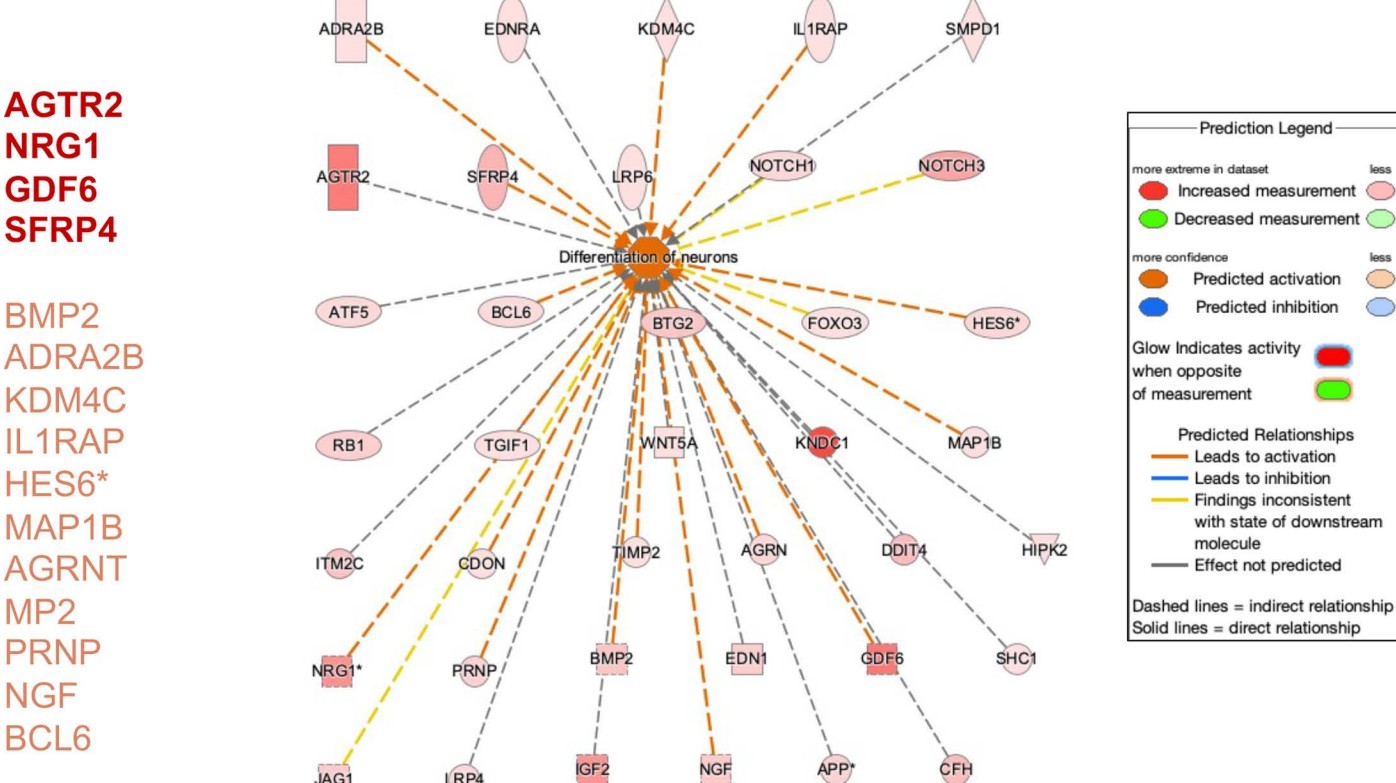

**Fig 3. Ingenuity pathway analysis shows significant overlap (p = 1.38E-68) between neuronal differentiation and upregulated differentially expressed transcripts in Arst/NI samples.** Differentiation of neuronal function is predicted to be activated (dark red color). Higher expressed genes are genes shown in darker red color.

and inhibits astrocyte development [34], is largely limited to brain and endocrine tissue expression. GDF6, another strong contributor to hippocampal neuronal differentiation is also linked to wnt-neural fate (Fig 4, circle 6). The prion protein gene (PRNP) was also elevated in Arst/NI cells, in accord with Prnp RNA upregulation previously identified by direct RT/qPCR analyses [4]. Finally, the NI neural transcripts appeared to be independent from the upregulated IFNs and inflammatory transcripts. Rather, neuron differentiation here was induced by the anti-proliferative transcripts.

**Arrest induces IFN stimulated and innate immunity responses in uninfected cells.**

Table 1 also shows strongly upregulated immune-associated complement factors Cfh (21 fold) and C1s (8 fold). In these monotypic neural cultures, there is no extrinsic source of complement RNAs so C1s must be endogenously produced. In addition, the glycoprotein clusterin, a chaperone and complement regulating factor was significantly elevated. Its elevation in NI arrested cells shows it is not linked to infection. The upregulation of Clu (clusterin) [35] and complement factors was unexpected. However, although not linked to neuron differentiation in the database, complement has been shown to be produced by neuronal cells in culture [36]. Analysis of the entire set of upregulated NI arrest vs proliferating transcripts further expands and highlights distinct transcript sets according to function. Downregulated transcripts were again dominated by anti-proliferative functions as shown in S2 Fig. Additionally, a small set of RNA processing features, including mRNA splicing, are known to be involved in selecting neuron specific RNA isoforms during neuronal differentiation. Upregulated transcripts fell into a large "anti-viral defense" set that contained innate immune functions (Fig 4, circle 1). Neuronal-linked

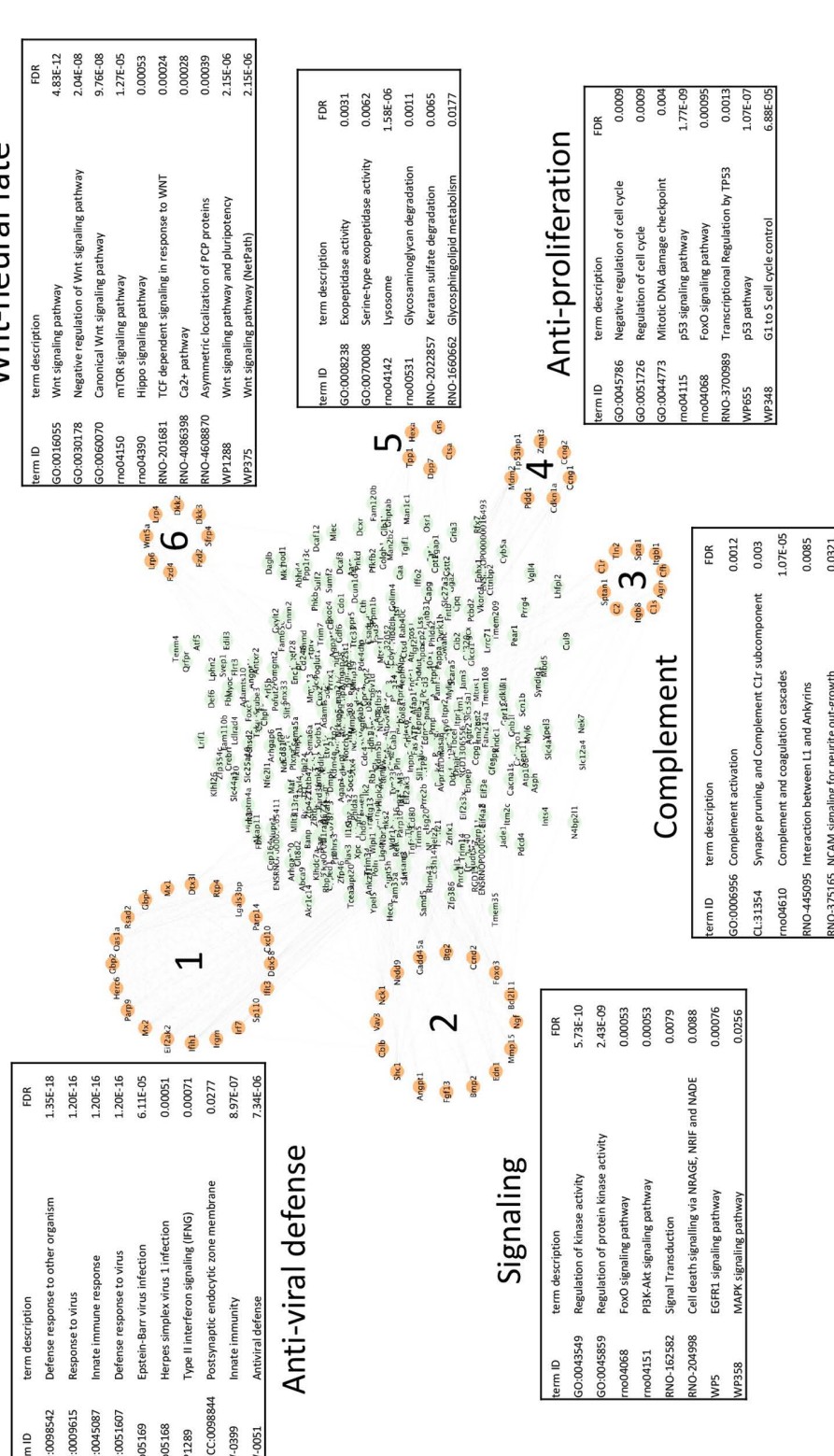

## Wnt-neural fate

| term ID | term description | FDR |
|---|---|---|
| GO:0016055 | Wnt signaling pathway | 4.83E-12 |
| GO:0030178 | Negative regulation of Wnt signaling pathway | 2.04E-08 |
| GO:0060070 | Canonical Wnt signaling pathway | 9.76E-08 |
| rno04150 | mTOR signaling pathway | 1.27E-05 |
| rno04390 | Hippo signaling pathway | 0.00053 |
| RNO-201681 | TCF dependent signaling in response to WNT | 0.00024 |
| RNO-4086398 | Ca2+ pathway | 0.00028 |
| GOCC:0098844 | Asymmetric localization of PCP proteins | 0.00039 |
| WP1288 | Wnt signaling pathway and pluripotency | 2.15E-06 |
| WP375 | Wnt signaling pathway (NetPath) | 2.15E-06 |

## Anti-proliferation

| term ID | term description | FDR |
|---|---|---|
| GO:0008238 | Exopeptidase activity | 0.0031 |
| GO:0070008 | Serine-type exopeptidase activity | 0.0062 |
| rno04142 | Lysosome | 1.58E-06 |
| rno00531 | Glycosaminoglycan degradation | 0.0011 |
| RNO-2022857 | Keratan sulfate degradation | 0.0065 |
| RNO-1660662 | Glycosphingolipid metabolism | 0.0177 |
| GO:0045786 | Negative regulation of cell cycle | 0.0009 |
| GO:0051726 | Regulation of cell cycle | 0.0009 |
| GO:0044773 | Mitotic DNA damage checkpoint | 0.004 |
| rno04115 | p53 signaling pathway | 1.77E-09 |
| rno04068 | FoxO signaling pathway | 0.00095 |
| RNO-3700989 | Transcriptional Regulation by TP53 | 0.0013 |
| WP655 | p53 pathway | 1.07E-07 |
| WP348 | G1 to S cell cycle control | 6.88E-05 |

## Anti-viral defense

| term ID | term description | FDR |
|---|---|---|
| GO:0098542 | Defense response to other organism | 1.35E-18 |
| GO:0009615 | Response to virus | 1.20E-16 |
| GO:0045087 | Innate immune response | 1.20E-16 |
| GO:0051607 | Defense response to virus | 1.20E-16 |
| rno05169 | Epstein-Barr virus infection | 6.11E-05 |
| rno05168 | Herpes simplex virus 1 infection | 0.00051 |
| WP1289 | Type II interferon signaling (IFNG) | 0.00071 |
| GOCC:0098844 | Postsynaptic endocytic zone membrane | 0.0277 |
| KW-0399 | Innate immunity | 8.97E-07 |
| KW-0051 | Antiviral defense | 7.34E-06 |

## Signaling

| term ID | term description | FDR |
|---|---|---|
| GO:0043549 | Regulation of kinase activity | 5.73E-10 |
| GO:0045859 | Regulation of protein kinase activity | 2.43E-09 |
| rno04068 | FoxO signaling pathway | 0.00053 |
| rno04151 | PI3K-Akt signaling pathway | 0.00053 |
| RNO-162582 | Signal Transduction | 0.0079 |
| RNO-204998 | Cell death signalling via NRAGE, NRIF and NADE | 0.0088 |
| WP5 | EGFR1 signaling pathway | 0.00076 |
| WP358 | MAPK signaling pathway | 0.0256 |

## Complement

| term ID | term description | FDR |
|---|---|---|
| GO:0006956 | Complement activation | 0.0012 |
| CL:31354 | Synapse pruning, and Complement C1r subcomponent | 0.003 |
| rno04610 | Complement and coagulation cascades | 1.07E-05 |
| RNO-445095 | Interaction between L1 and Ankyrins | 0.0085 |
| RNO-375165 | NCAM signaling for neurite out-growth | 0.0321 |

**Fig 4. Interaction network analysis of upregulated differentially expressed genes in ArsnI/NI vs ProI/NI.** Interaction network was built using STRING knowledgebase (v. 12.0). The resulting network was imported into Cytoscape (v. 3.10.2), and clustering analysis was performed with MCODE application (v. 2.0.3) using a degree cutoff 2, node density cutoff 0.1, K-core 2, and maximum depth 100. Resulting clusters were functionally annotated on STRING knowledgebase.

differentiation transcripts form two separate sets: A) with complement activation and neurite processing/synaptic elaboration (circle 3) and B) Wnt-Neural fate pathways (circle 6). Upregulated anti-proliferative pathways and less specific signaling activities are also seen. Together Figs 3 and 4 indicate innate immune transcripts separately contribute to neuron differentiation in uninfected SEP cells rather than any anti-viral response.

Ingenuity pathway analyses expanded number of innate immune transcripts linked to neuron differentiation. Many of these highly upregulated anti-viral transcripts are induced by IFNs, especially IFNγ in S3 Fig showing strong upregulated viral responsive transcripts including TLR3, TRIM14, RTP4, RSAD2, ISG20, OAS1, GBP4, CD80 and MMP13 as well as multiple weaker IFN-induced transcripts. The extent of this group further verifies the upregulation of β-IFN found previously by direct RT/qPCR where OAS was upregulated 8-fold during 48–60 fold elevations of β-IFN [4]. It also underscores many additional IFN stimulated transcripts in Arst/NI cells including IFN-γ, IFN receptor (IFNar), Isg20 and others including upregulated Ifr7 (see complete deposited dataset). Gene Set Enrichment analysis in S4 Fig further solidified the strong upregulation of IFNA, IFN-B1 and STAT 1 targets, with corresponding downregulation of EGFR and G2-M cell cycle signaling. DNA methylation was also downregulated during arrest, consistent with enhanced transcriptional activity of the neural differentiation transcripts identified, rather than a detrimental cell stress or toxic change.

In sum, physiological arrest induced a network of robust IFN responses unrelated to infection that are typically linked to innate immunity. The arrested cells displayed no cytopathic or toxic effects and remained in an arrested state for many days. The interferon network revealed above appears to be integral to the differentiation program at least for some neuronal types, and complement, typically associated with innate immune and anti-viral responses, appears to participate positively in this process. By comparison CJ infected cells, that could not be superficially distinguished from uninfected cells in the proliferative state revealed major global transcript differences affecting proliferative pathways, neural differentiation and several innate immune responses.

### Arrested CJ+ versus proliferating CJ− transcripts

As shown in Fig 1C prior CJ infection dramatically changed the latently infected CJ− cell phenotype as evidenced by the reduced scale and range of transcripts that can be elicited by arrest. Most differences in this comparison showed up as a failure to inhibit DNA replication and cell cycle checkpoint pathways in arrested CJ+ cells. For example, 23 different histone transcripts were upregulated in both sets of CJ+ cells (at 6 and 34 days with 2 and 10 TCID/cell respectively). This coincided with limited ongoing cell division. In contrast, these histones were downregulated in Arst/NI samples (S1 Table). The significant downregulation of the ABRA signaling pathway that conveys external signals, including serum signals, further shows CJ+ cells escape from the low serum environment. Either CJ− cells were imprinted by previous infection, priming them to escape or resist arrest. Alternatively, recrudescence of high infectious agent titers during arrest activated proliferation pathways. Upregulated proliferative transcripts in CJ+ cells were expected because CJ+ cells had to be split 16–18 days after arrest whereas uninfected cells remained in a non-proliferative state for 27 days [4]. The extent and number of proliferative activated transcripts identified here were very large and not anticipated.

### Minimized neuronal differentiation in CJ+ cells

Combinatorial comparisons underscored more granular differences including a diminished activation of neuronal transcripts in CJ+ cells. Fig 5 compares Arst/CJ+ versus Prol/NI and Prol/CJ− cells and segregates common and unique cDNAs in uninfected and CJ+ arrested samples. A reduced number of strong neuronal differentiation transcripts is apparent in Arst/CJ+ vs Prol/CJ− sets. Cavin4, an important component of synaptic caveolae, is also upregulated in NI cells but downregulated in all CJ cells, including proliferating CJ− cells. This underscores a permanent imprinted change in latently infected CJ− cells that limits neuronal development. In contrast, Wnt (Wnt5a) signaling is upregulated in both Arst/NI cells (Fig 4) and CJ+ cells (S5 Fig). This upstream transcript positively regulates neuron maturation, synaptogenesis and

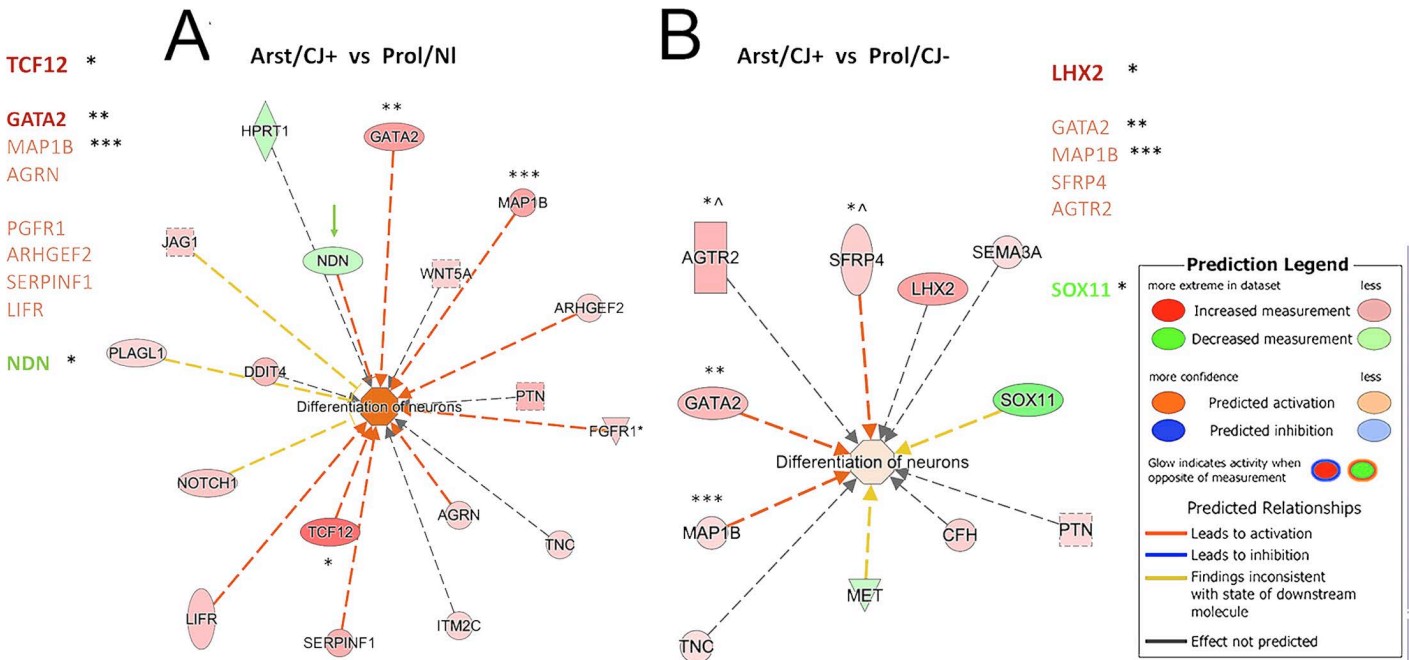

**Fig 5. A) Ingenuity pathway analysis.** Significant overlap (and activation) between differentially expressed genes and neuron differentiation in Arst/CJ+vs Prol/NI cells **(A)**, but it is reduced as compared to stronger activation in Arst/NI vs Prol/NI transcripts (see Fig 3). The differentiation of neurons is even less activating in **B**) that shows Arst/CJ+vs Prol/CJ− differences. Note the pale pink center in B with the darker red neuronal activation in A. The transcripts are also different in each of these CJ+ comparisons with only Map18 shared (***) in all 3 comparisons. GATA2 was activated only in these two CJ+ comparisons, but not in Arst/NI cells (**), and two stronger transcripts (TCF12 and LHX2) were found only in a single comparison (*). Strong NGR2, GDF6 & SFR4 signals seen in Arst/NI samples (see Fig 3) are absent in these CJ+ comparisons and strong NI AGTR2 is absent or diminished in the CJ+ comparisons above.

axonal and dendritic outgrowth, and neuritic outgrowth is observed in NI and CJ arrested SEP cultures. Together, this data suggests that the CJ agent imprints selected downstream synaptic and neuronal elaboration components and pathways to diminish neuron-specific functions.

Combinations of different sets of uninfected and infected samples further expanded major differences in CJ+ and NI cells as seen in Fig 6. The highest z scores in the Arst/NI vs Prol/NI comparison show strong downregulated (blue) pathways. In contrast, the highest z scores in Arst/CJ+vs Arst/NI are upregulated genes that enhance DNA synthesis and cell division, or that have unrelated functional changes. Moreover, five of these same pathways (dotted in red) show opposite regulation between the two sets  (Fig 8 2 top sets). These examples indicate the depth and range of escape from anti-proliferating signal in CJ+ cells. To complete the comparisons for CJ+ changes, the bottom two sets in compare Arst/CJ+ to itself (Prol/CJ−) and to Prol/NI. These comparisons highlight additional major differences between NI and CJ+ samples. Notably, the highest CJ+ scores included hyperchemokinemia, caused by viral and other pathogens, cytokine storm signaling transcripts, and proinflammatory Il17 upregulation that promotes and exacerbates viral induced effects. These are not present in the comparable NI control set with lower upregulated z-scores  (top left). Degradation of extracellular matrix and Matrix Metalloproteinases are also high in this CJ+set but absent in the NI set. These CJ+ differences indicate their strong link to high titers of agent rather than to IFN related changes common to both NI and CJ+ arrested cells.

## 22 transcripts activate anti-viral activity in CJ+ cells

Ingenuity pathways yielded additional immune and anti-viral distinctions between the Arst/CJ+ and Prol/NI sets. These included both up (red) and down (green) regulated RNAs as indicated by their blue lines in. Major strong regulators are

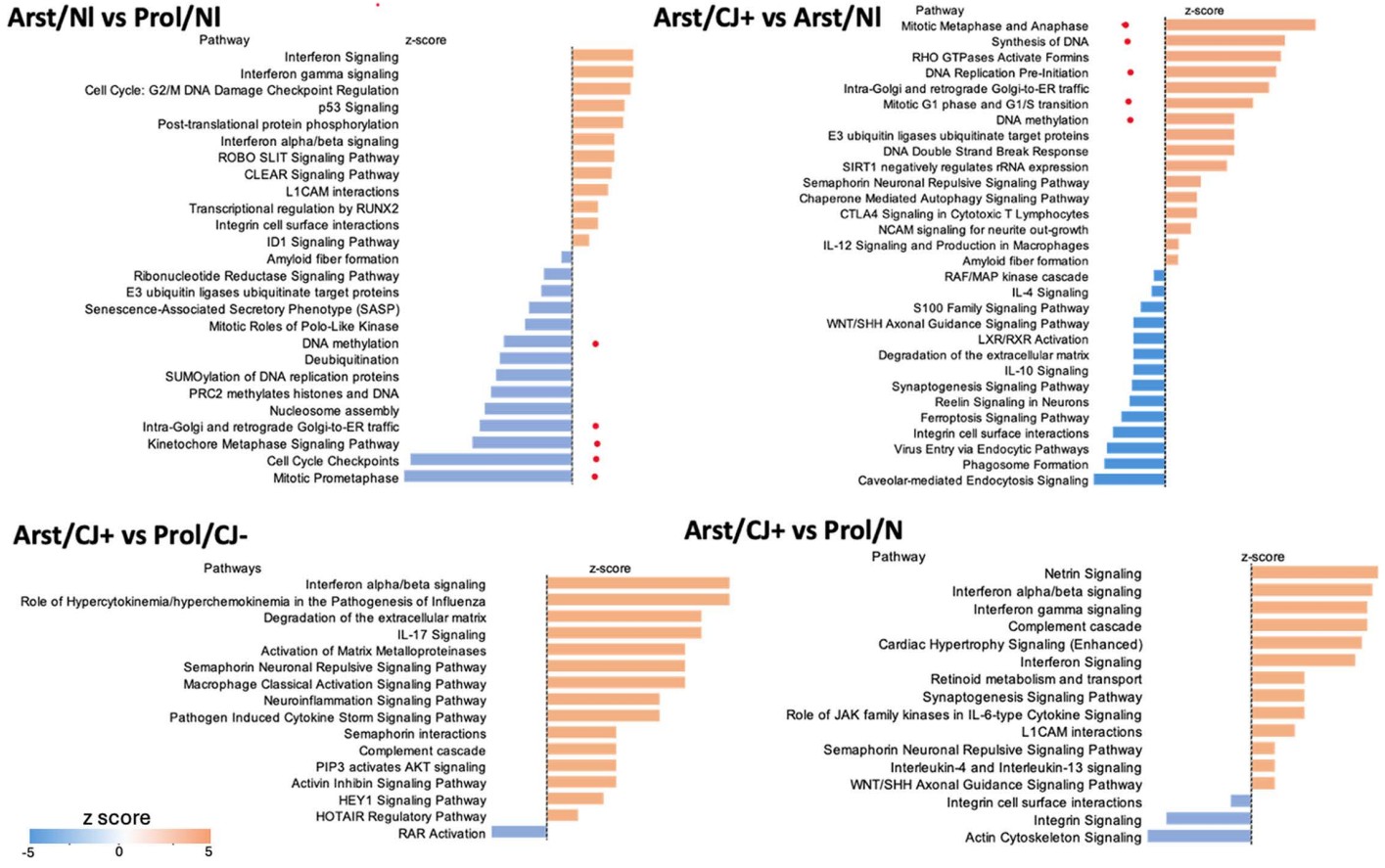

**Fig 6. Ingenuity pathway analysis showing overrepresented pathways (Fisher's exact test FDR p<0.05), and activation status prediction based on z-score for the comparison of different sets of infected and uninfected cells.** Positive z-score (orange bars) indicates pathway activation while negative z-score (blue bars) indicate inhibition.

listed. The strongest Mx1* gene, induced by IFNs, antagonizes the replication process of several different RNA and DNA viruses. Antiviral HERC6, ISG15, Itgb3 and IFIT 3 were all identified as strong activators in CJ+ cells as shown in Fig 7. This anti-viral response also revealed a strong interactive network with the IFNs upregulated in the CJ+ dataset and is shown with cellular sites of action (S6 Fig). In CJ+ cells the IFNs are not extracellular in origin but made by the cell since no other cells are present to produce activating cDNAs. This data again substantiates RT/qPCR direct experimental results in CJ+ cells: IFNs were far higher than in uninfected cells, indicating IFN enhanced transcripts were activated by the recrudescence of productive infection.

#### Stronger innate immune and anti-viral patterns in arrested CJ+vs NI cells

IPA graphic analysis of signal strength in comparative NI and CJ+ sets further solidified the above findings as shown in Fig 8. Panel A shows 11 of 16 upstream regulators produced in CJ+ cells (red) are not present in Arst/NI cells. Panel B shows selected canonical pathways with pathogen induced hyperchemokinemia, cytokine storm, matrix metalloproteinases and Il17 in CJ+ samples. These were not upregulated in Arst/NI cells.

Unique pathways were further interrogated in both CJ+/NI and CJ+/CJ− datasets, shown in S7 Fig, where plots of additional IPA transcript differences are apparent in both CJ+ comparisons. These transcripts further accentuate viral hypercytokinemia (not present in Arst/NI vs Prol/NI comparisons), stronger IFNα/β signaling, rheumatoid arthritis signaling,

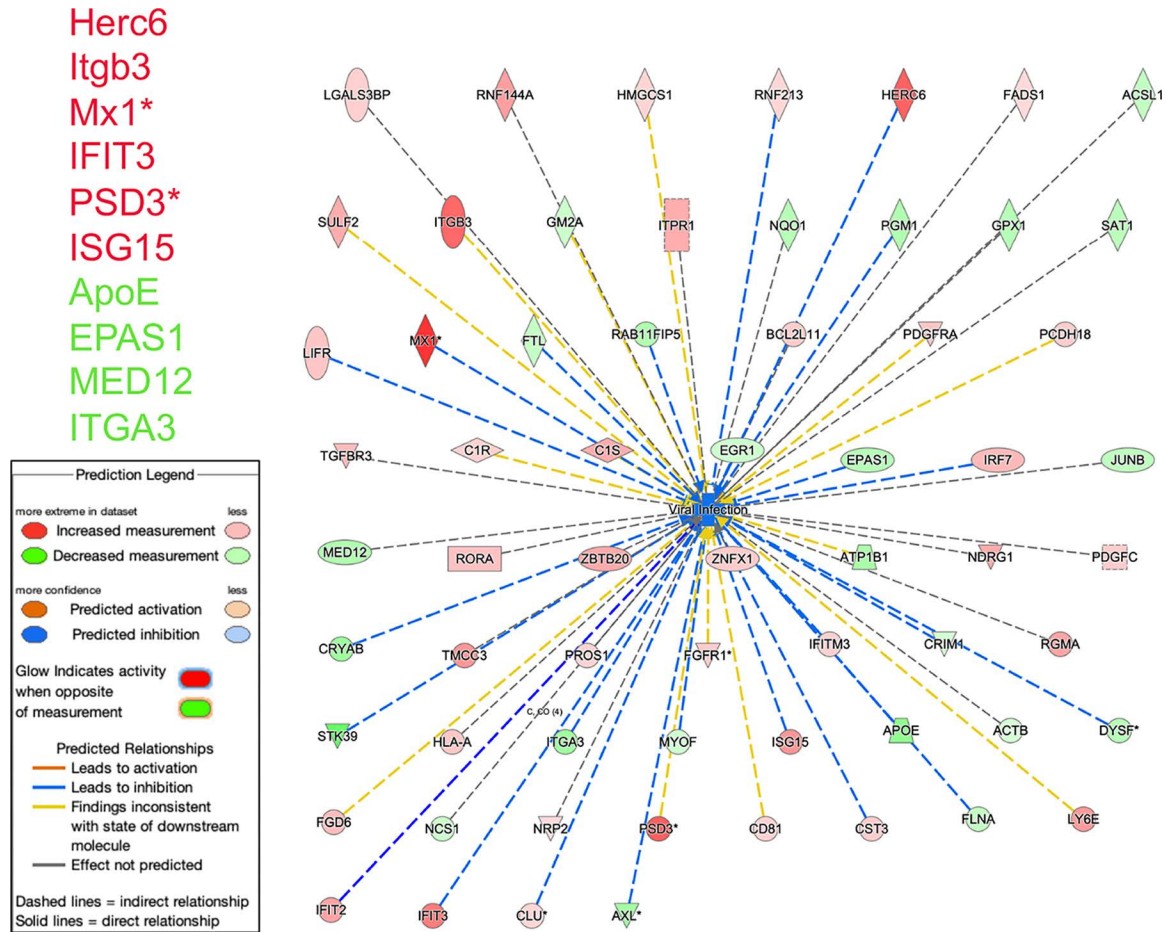

**Fig 7. Ingenuity pathway analysis overlap between differentially expressed genes and viral infection function in Arst/CJ+vs Prol/NI.** Strong anti-viral activation (blue lines) from transcripts up regulated in CJ+ cells. Mx1 gene protects against viruses that replicate in the nucleus, possibly indicative of a CJ+DNA component. Although PSD3 (plectin) is captured here by ingenuity viral pathways (without a consistent role), it is highly expressed in neurons but was not IPA linked to neuronal function or differentiation (see Fig 3).

metalloproteinase activation, degradation of extracellular matrix and O-linked glycosylation that are not present in Arst/NI samples. Thus, during arrest, infected cells recruit an enhanced and more diverse immune response than that found in differentiating normal cells. Interestingly, several of these increases were not seen in CJ+vs CJ− sample comparisons. This again underscores and expands the underlying changes in CJ− cells imprinted by prior infection that were brought out during re-arrest.

Arrested NI cells unlike CJ+ cells also displayed robust senescence pathway transcripts in addition to TP53 RNA signaling which regulates transcription of cell death genes. In this case senescence was linked to observed neurodifferentiation rather than cell death or degenerative changes. The widespread down regulation of proliferative elements in NI cells, but not CJ+ cells, was also evaluated by Gene Set Enrichment Analysis (GSEA) shown in S8 Fig. These GSEA analyses of CJ+vs Arst/NI also revealed strong upregulation of α and γIFN targets in addition to increased methylation of anti-proliferative (arrest) transcripts, the opposite of the downregulated methylation (activation) of anti-proliferative transcripts in NI cells. Methylation is a major regulator of stable epigenetic changes and methylation of cell cycle checkpoints may be one of the mechanisms that permanently imprint the increased proliferative activity observed in CJ+ cells. CJ+ cells also show strong downregulation of clathrin endocytosis pathways and Cavin4 synaptic vesicle formation, both of

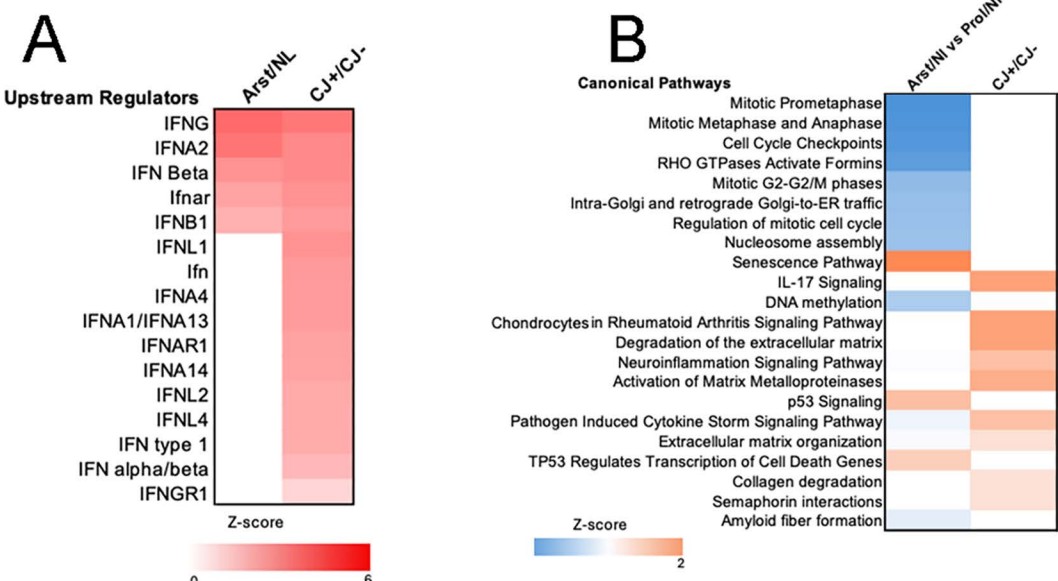

**Fig 8. Ingenuity pathway analysis of upstream regulators.** Comparison of upstream regulators and examples of canonical pathways that differ in Arst/NI and CJ+ cells. A) shows activation status (based on z-score) of expressed upstream IFN regulators (red) not present in Arst/NI cells. B) shows canonical pathways. Note other signature transcripts upregulated (orange) in CJ such as extracellular matrix regulation. DNA methylation and mitosis are downregulated (blue) in NI vs CJ.

which are reported to be central for PrP processing [37,38]. Neither downregulated genes affected the steep rise in infectivity. The more central and major player in TSE infections, Prnp, was also downregulated 4 fold in Arst/CJ+ cells versus Arst/NI cells. This RNA-seq data corresponds well with the more specific 5 fold RT/qPCR transcript downregulations previously reported. Moreover, corresponding PrP and PrP amyloid reductions of 3–5 fold were also documented in different passages [4] during PrP infection of naïve cells. This is notable because naïve infected cells showed no decrease in PrP. Downregulated Prnp in CJ+ re-arrested samples represent only 1/341 unique CJ+ changes by Venn diagram that encompasses a complete list of unique CJ+ transcripts S9 Fig.

Unique CJ+ vs Arst/NI cells show both upregulated (n = 195) and downregulated (n = 146) transcripts. This more complete data further supports many of the consolidated limited observations depicted above. However, additional transcripts, such as a few upregulated tumor suppressors and neuronal differentiation genes were not picked up by directed queries of IPA analyses, possibly because they have not yet been linked in that database to either neuronal differentiation or inflammation. These additional transcripts could signify a more robust complex network of immune/neuronal differentiation changes. This more detailed supplement also shows the extensive fundamental difference in CJ+ versus other groups. The limited response of proliferating CJ− versus NI cells indicates even more profound hidden changes may be induced during prior latent infection. There are only 40 upregulated unique changes induced in arrested CJ− cells (pink column) while there are 86 unique changes in the NI comparison (light green) as seen in S10 Table. This extensive list can easily be searched for other genes of interest in TSEs, as for example Adamts10 that acts at the cell surface in collaboration with PrP. Downregulated genes show the same pattern, with only 18 differences in CJ− versus 26 in NI comparisons. Other unique transcripts identified here that may participate in CJ latency and pathology can now be tested in different cell and animal disease models. In sum, cDNA libraries demonstrated a multitude of previously unrecognized innate immune and anti-viral pathways activated by physiological arrest. Uninfected cells displayed a robust recruitment of neural differentiation transcripts induced during perpetual arrest, along with a surprisingly wide variety of IFN and innate immune pathway responses. These pathways and transcripts were reproducible in independent experiments and appear to be integral to

progressive differentiation and not due to underlying cytotoxic or added inflammatory stimuli. Arrest induced changes in latently infected CJ− cells showed marked differences from arrested uninfected control cells in proliferation, neuron differentiation and extent of innate immune responses. Since these CJ+ changes were not induced by arrest of uninfected cells, they are ultimately caused by infection. Of the many different transcript changes uncovered here, the most dramatic and far-reaching change was the release from proliferation controls in CJ+ cells. These multiple transcripts underlie the observed partial escape from arrest as compared with the complete arrest of uninfected cells. At the same time, CJ+ cells suppressed the neuron differentiation program of normal cells. These CJ+ differences strongly suggested a divergent phenotype imprinted by prior and/or latent infection as discussed below.

In sum, Fig 9 shows a graphical abstract of the experimental strategy, comparative data analysis and major reproducible RNA-seq conclusions. As previously shown, major biological proliferative changes were expanded by RNA-seq. Consequences of proliferative arrest were followed by significant neuronal differentiation and immune transcript responses. In comparison with normal uninfected neurons, productively infected CJ+ cells subverted proliferative controls and neuronal differentiation while they enhanced innate immune responses. Moreover, RNA-seq shows that CJ infected cells can remain in a prolonged non-productive latent state, yet they retain imprints of previous productive infection.

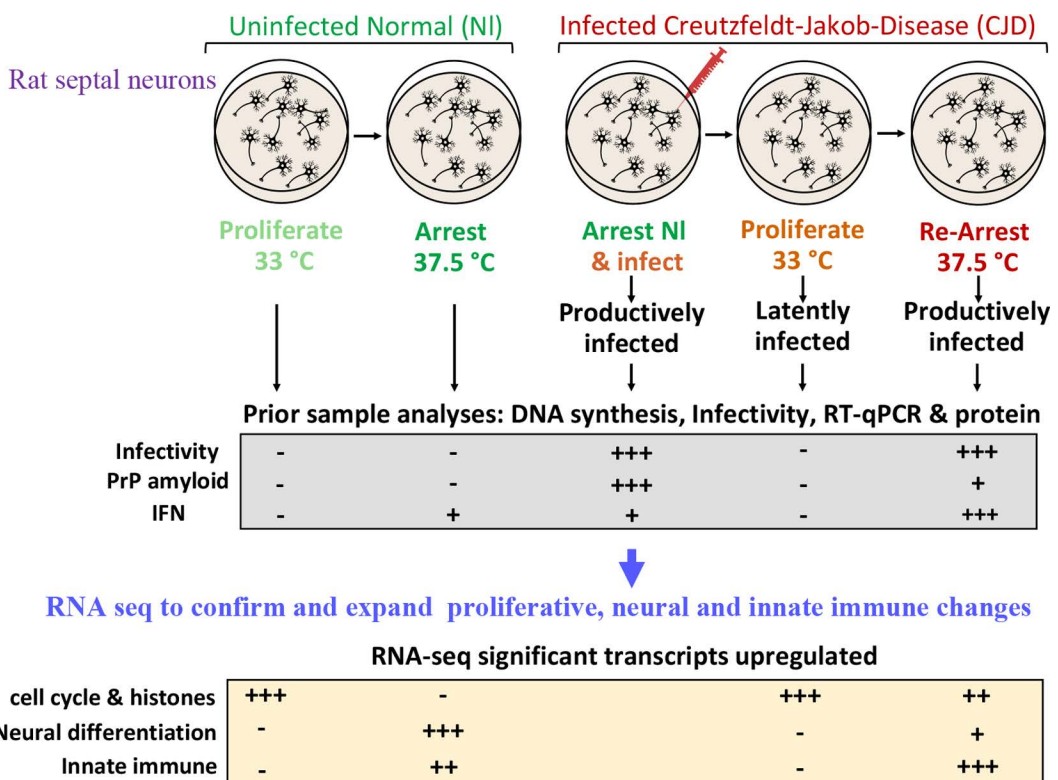

**Fig 9. Graphical summary: Normal rat septal neurons, engineered to reversibly proliferate and arrest DNA synthesis, were compared with CJ infected cells (top row).** Previous DNA experiments, RT/qPCR analyses, prion protein (PrP & PrP amyloid) evaluations, and physiological changes are summarized in the middle row. These finding were confirmed and expanded by RNA seq (bottom row). Numerous major transcriptional differences in normal as compared with latently infected and productively infected neurons were revealed. Abundant cell cycle and histone transcripts controlled normal arrest. This arrest was followed by neuronal differentiation along with upregulated innate immune transcripts. The same conditions of arrest in CJ+ cells subverted cell cycle and histone controls and neural differentiation whereas innate immune responses were enhanced (+++) while – is no change.

## Discussion

The ultimate cause(s) of human neurodegenerative diseases, such as Alzheimer's Disease, are difficult to ascertain because critical insults can occur years before symptoms become apparent. Neurons are at the center of pathologic and functional changes while innate immunity can induce latency or accelerate disease progression, especially in hidden or non-productive viral infections. Because CJD can be latent for 38 years, as in CJD contaminated growth hormone cases, we developed a unique model of latent non-productive infection in neurons that contrasts with shorter latent productive brain infections. Proliferating and arrested neurons were engineered to reversibly differentiate; at 33°C they proliferate but at 37.5°C they are arrested. Arrested normal cells produced a plethora of anti-proliferative RNA transcripts as would be expected from the documented lack of DNA synthesis in arrested neurons. Additionally, known neuron differentiation transcripts, including Prnp were upregulated along with many innate immune responses, including cytokine/chemokine transcripts not previously associated with neuronal differentiation.

To generate latently infected CJ− cells, arrested high infectivity (CJ+) cells were reverted to a proliferative state at 33°C and showed no detectable infectivity (CJ−). When these CJ− cells were re-arrested they rapidly developed high infectivity (CJ+), proving they were latently infected. CJ+ neurons downregulated many replication controls, suppressed neuronal differentiation transcripts including Prnp, and enhanced IFN stimulated pathways and anti-viral transcripts in accord with their very high β−IFN RNA transcripts by previous RT/qPCR assays. None of these numerous changes limited productive infection. Latent non-productive CJD infection in these cells also epigenetically imprinted many proliferative pathways to thwart complete arrest. The following discussion focuses on a detailed exposition of the contribution of specific transcripts and pathways that support the global biologic changes observed here.

Library cDNA construction with bioinformatics analysis uncovered previously unsuspected actors and global changes that could not be revealed by limited individual RT/qPCR studies. The data here expands the range of centrally involved pathways for proliferative control, neural differentiation, and IFN stimulated and other innate immune transcripts. In normal uninfected cells, conditional physiological arrest induced a total of 379 differentially expressed genes, and the overwhelming majority of the 204 downregulated transcripts suppressed DNA replication. Many upregulated anti-proliferative transcripts further targeted a diverse set of cell cycle checkpoint and mitotic cell transition controls, e.g., by inhibiting G1/S phase and mitotic transitions. These proliferative inhibitory changes were remarkably diverse and virtually complete by 6–7 days and were total by 14 days post arrest in all 3 independent experimental samples collected at 3 different times.

Arrest of normal cells also upregulated known neuronal differentiation transcripts. Ingenuity pathway analysis highlighted 4 strong neuronal transcripts (AGTR2, NRG1, GDF6 and SFRP4) in addition to 11 other neural transcripts including neuregulin, Prnp and HES6. HES6, a helix-loop-helix transcription repressor protein, is mainly transcribed in brain and endocrine tissues and this transcript can participate in epigenetic modifications of proliferation and neurodifferentiation. Arrest also downregulated methylation, another mechanism for epigenetic activation of neuronal transcripts. The basic cell fate of SEP cells was already imprinted when they were engineered to proliferate, and stable neural differentiation transcripts were documented here post arrest. In cultured differentiated post-mitotic cerebrum neurons, stimulus induced changes evaluated over several days revealed selective intron retention in the nucleus linked to neurodifferentiation [39,40]. Stable nuclear intron change may also be informative for the widespread epigenetic anti-proliferative and innate immune responses brought out by arrest of normal SEP cells.

Latently infected proliferating SEP cells (CJ−) could not be distinguished from uninfected NI proliferating cells by their behavior or morphology. Remarkably, these CJ− cells continued to differentially express several relevant transcripts. This indicated that previous and/or latent nonproductive infection was sufficient to change and permanently imprint a CJ altered phenotype. These unique transcripts in CJ− proliferating cells included upregulated acute phase protein LPB, complement (C1s), and Mt1M, an anti-inflammatory molecule [41] along with downregulated Cavin4 and neurodevelopment genes Tenm3 and Dpys13. All of these were indicators of subsequently enhanced inflammatory responses, along with suppressed neurodifferentiation, in re-arrested CJ+ cells.

Re-arrest also brought out a variety of imprinted transcriptional proclivities in CJ− latently infected cells. The data here were reproducible in independent culture passages of cells re-arrested at different times up to 34 days. In contrast to NI cells, high titer arrested CJ+ cells suppressed many NI proliferative controls and enhanced innate immune cytokine/ chemokine and inflammatory transcripts. For many years TSE agents have been considered immunologically silent, largely because they elicit no lymphocytic infiltrates that are typically induced by acute and subacute viral infections. Nor have neutralizing antibodies been found. However, innate immune pathways in CJ+ were significantly enriched compared to NI arrested cells. Remarkably, the CJ+ ratio of immune to proliferative changes was the reverse of arrested NI transcripts, with innate immune pathways more upregulated while proliferative pathways were downregulated by infection (see Fig 6). CJ+ cells also enhanced immune transcripts during the same time that the neuronal differentiation program was suppressed. This indicated that the enhancement of innate immune transcripts in CJ+ was not dependent on neuron differentiation. Previous direct RT/qPCR studies of β−IFN were also validated and expanded in CJ+ cells and included upregulation of many IFNα and IFNγ targets and cytokines. In acute viral infections, such as with Semliki virus, the response of immature neurons is augmented byβ−IFN which reduces acute viral replication [42]. In contrast to Semliki virus, multiple anti-viral transcripts and IFNs failed to stop rapid CJ agent replication in SEP neurons. We are unaware of any form of recombinant PrP or PrP amyloid that upregulates the IFNs or antiviral transcripts identified here, and the enhanced CJ+ anti-viral and IFN activated transcripts observed here may be, at least in part, directly caused by the infectious ~20nm particle [6,43], or indirectly signaled, as by agent induced dsRNA. Regardless, enhanced innate immune and anti-viral responses coincide with productive infection.

The 4-fold reduction of Prnp transcripts in CJ+ cells also correlated well with previous RT/qPCR and PrP analyses [4]. Although host PrP is essential for CJ infection, the reduction in Prnp and PrP amyloid found here may be part of a host defense mechanism imprinted during prior CJ infection. Importantly, naïve SEP cells that were infected previously did not show a reduction in PrP unlike re-arrested SEP cells. Instead, they attained the same 8–10 fold increased PrP as their NI counterparts [5]. This data further supports a retained epigenetic imprint in latently infected CJ− cells. This CJ imprinted downregulation of Prnp, PrP, and PrP amyloid, in addition to downregulated Cavin4, failed to suppress high levels of infection. Cavin4, the caveolar protein in membrane lipid rafts, has been claimed to be required for the proposed conversion of normal host PrP into its misfolded, presumably infectious scrapie amyloid form [38].

Although neurons were originally thought to be the only cells with high infectivity, possibly because they show high levels of PrP amyloid, CJ brain infection activates myeloid microglia, and isolated living microglia carry infectivity titers equivalent to whole brain homogenates [44] even though they contained barely detectable PrP and no detectable PrP amyloid. Isolated microglia also displayed several IFN regulated and activated genes, e.g., IFI204, IRF2 and IRF8. Several additional immune and inflammatory responses were brought out by poly I:C (e.g., ISG15, CXCL10 and OAS) stimulation [45], and they correspond to the CJ+SEP transcripts identified here, suggesting a common response to infection by very different cell types. Moreover, in CJD infected mouse brains, the IFN linked transcripts IFI204, IFI 202, CXCL10 and IFR8 by RT/qPCR were elevated as early as 10–40 days post-inoculation, a time of progressive agent exponential doubling [46]. PrP amyloid becomes detectable only 50 days later when infectious titers have risen by 5–6 logs. This very early period of latent infection is rarely studied, and the SEP cell model here shows how critical conditional changes are for productive infection.

Progressive FU-CJD intracerebral inoculation studies [46] also revealed that cytokines TGFα, IL1β, β chemokines and MIP-1α were elevated ≥10-fold within 30 days of brain inoculation. Since none of these changes were present in mice inoculated with uninfected brain, they represent an important recognition by the host of the infectious agent. A similar separation of infectious agent and PrP amyloid effects is seen in progressive rat disease; activated microglia appear midway in the incubation period, and it takes 100 days more for PrP amyloid with spongiform changes to become detectable [8]. This further substantiates host innate immune responses to the CJ agent are unrelated to PrP amyloid and neurodegeneration.

While little is known about latency in peripheral cells that are first targeted via natural routes of infection, e.g., via skin abrasion (endemic scrapie) and oral (epidemic BSE) routes, TSE experimental studies clearly show myeloid and lymphoid cells harbor infection for extended times, especially with low dose infections that may not be manifest in brain during an animal's normal lifespan. During this latent period RNA-seq can diagnostically exploit shared immune-linked pathways, especially in accessible infected peripheral immune-related cells such as dendritic gut cells that are infected long before brain [47], and in tonsil and appendix. While infection activated molecules may vary in different cell types and species, their comparable immune-linked functions and overlapping pathways appear to be generalized in a variety of TSE infections (agent-strains and species).

Unlike cell cultures, animals have many additional cellular and humoral ways to hide and restrict TSE agent replication as evidenced by rapid TSE agent replication after infectious transfer from mice to murine cell cultures and visa versa, e.g., [19]. A variety of TSE agents consistently doubles every 19–24 hours in GT1 neuron derived cultures whereas dramatically longer clearly different 5–25 day agent doubling times are found in mice inoculated intracerebrally [23,48]. These findings underscore the release of the infectious agent from complex host controls. When infected cell material is reinoculated in mice these agents again show a vastly slower replication and agent-strain specific regional brain changes. Many different host responses, including common lysosomal clearance mechanisms, and PrP amyloid itself, can eliminate infection in culture [9], but effective inhibitors of TSE agent replication in animals remain largely uncharted, especially in peripheral lymphoid tissue where these agents silently persist for many years.

The multiple modified proliferative transcripts in CJ infected SEP neurons were also remarkable. Re-arrested CJ+ cells subverted proliferative controls as documented by previous cell proliferation studies, and accordingly was the most profound and global change uncovered by RNA-seq. This led to a partial but permanent escape from arrest in CJ+SEP cells that had been previously observed biologically. Increased cell replication is part of the fundamental biology of TSE infections in both scrapie and CJD and is not limited to neuronal cell types. Although largely forgotten, faster growth of explanted scrapie and CJD infected brain cells is reproducible and has been reported by at least 4 other independent laboratories [49]. Rapidly proliferating permanent cell lines that continue to produce infectivity in addition to those that have lost infectivity were identified. Several of these cell lines showed a transformed immortal phenotype with loss of contact inhibition. Others formed huge tumors on heterologous transplantation to nude mice. These tumors originated from sCJD human brain cultures and sCJD infected hamster cells [50]. This transformation feature, combined with a prolonged non-productive latency, is typical for transforming DNA viruses.

Although there is widespread belief that misfolded PrP amyloid is infectious without any nucleic acid [51], infection by recombinant PrP amyloid ("the gold standard of proof") has not been reproducible, and 20,000 recombinant PrP (recPrP) experiments showed no infection by validated assays, e.g., [52–54], A recent review of an additional 22 recPrP infectivity reports from other laboratories [13] shows the majority failed to produce infection from misfolded recPrP although they did show brain PrP aggregates and PrP "abnormalities" that were not transmissible. While the search for a specifically folded recPrP structure that is infectious continues, many investigators now consider additional component(s) are needed for infection. Nucleic acids would seem likely candidates since they have the capacity to define different stable TSE agent-strains.

While no TSE specific nucleic acid sequence has been delineated, highly infectious ~20nm infectious TSE particles have been isolated from brain, and these particles do not bind PrP antibodies and correspond to those seen ultrastructurally in N2a and GT1 cultured neurons [6]. Gradient isolated infectious particle fractions contain mitochondrial genome contamination in addition to circular DNAs of 1–3kb with phage linked REP sequences [55,56] and a REP protein appears with neurodegenerative synaptic changes in CJD brains [57]. While only limited nucleic acid sequences have been investigated, an essential nucleic acid component of TSE infectious particles cannot be excluded because nucleases that digest nucleic acids destroy 3 logs of infectivity in both CJ and scrapie infectious particles while PrP-amyloid is preserved [14].

The present SEP CJ model is unique because it shows an extended non-productive latency that mimics long non-productive natural and iatrogenic peripheral infections, including CJD human growth hormone infections. We examined only a single human TSE agent-strain; other TSE agents might not show an identical pattern of responses. However, the fundamental biology of all TSE agents (modes of agent spread, latency, and basic brain pathology) are common to all TSE infections, and different agent-strains are likely to show only minor differences in the degree of change. Arrest caused far more complex and abundant changes in CJ+ compared to uninfected NI cells. CJ− cells were primed to respond differently to arrest than NI SEP cells, and they retained a cellular memory of prior infection. Although the mechanism(s) of imprinting is not clear, methylation patterns were distinct in CJ versus normal SEP neurons. Additionally, others have reported human sCJD and sCJD infected mouse brain transcripts reveal an "epitranscriptomic" profile with altered RNA edited pathways [58]. Such profile changes, along with retained nuclear introns [40], might also underlie episomal changes in SEP neurons. In any case, accessible latently infected spleen and lymphoid myeloid cells may be conditionally manipulated by stress or arrested by drugs in culture to elicit productive infection. Resulting biological and transcriptional changes such as increased proliferation, or activation of unique transcript patterns that can be rapidly assayed by RT/qPCR, could facilitate diagnosis of latent infection. Some of the unique transcripts induced by CJ infection can also be of use in such studies.

The current results bring to the fore many common biologic, virologic and episomally altered pathways that may be diagnostically exploited. While previous experimental studies have shown productive infection occurs during prolonged non-clinical infections of brain, the above studies show TSEs agents can persist in a non-productive state that can be activated physiologically. This is relevant for human CJD infections that have a latency of years, a situation that parallels many other established latent infectious agents from tuberculosis to shingles and other viruses. White blood cells, intestinal and other tissue myeloid cells can disseminate TSE agents to the brain even though they may not be replicating detectable levels of infectious particles [17,44,47]. Finding and eradicating these cells is critical for preventing progressive brain disease.

## Conclusions

Post mitotic neurons are imprinted with a neuronal cell fate. Their neuronal differentiation program here was suppressed when they were reverted from an arrested to a proliferative state via a Ts SV40 T antigen construct. Physiological arrest induced cessation of DNA synthesis and cell proliferation. In uninfected SEP neurons arrest induced a plethora of cell cycle and DNA synthetic transcripts to control proliferation. Sequential time studies demonstrated simultaneous activation of transcripts for neuronal differentiation, IFNs and their activated innate immune pathways in accord with previous RT/qPCR assays.

Proliferating neurons with latent CJ infection showed rare transcriptional differences from normal proliferating neurons. However, on re-arrest many normal proliferative cell cycle and DNA synthesis linked transcripts were undermined. Orderly progression of programmed neuronal differentiation was also subverted by infection whereas innate immune viral-linked transcripts were enhanced. These changes took place before the highest CJ infectious titers (10 infectious particles/cell) were produced at 34 days. Thus, prior CJ infection can modulate and/or epigenetically sculpt nuclear transcription.

New conclusions here are: 1) Normal neuronal differentiation can incorporate innate immune responses; 2) Productively infected CJ+ cells escape arrest by subverting numerous histone and cell cycle controls, and they strongly upregulate multiple anti-viral pathways including IFNs and cytokine/chemokines; 3) These agent-induced changes are not phenotypically apparent and/or directly related to PrP or PrP amyloid; 4) The latent cell model of infection here is directly relevant for the years of covert human and animal TSE infections that may be activated by stress, immune changes and senescence (aging). Similar triggers may elicit progressive Alzheimer's Disease.

## Supporting information

**S1 Table. Top 25 genes down regulated in each of 3 independent samples from different passage of NI cells, i.e., Arst/NI for 7 days from passage 7, at 6 days from p 25, and at 14 days from p25.** All genes are downregulated in at least 2 of the 3 samples (see text) with comparable fold changes (Log$_2$Ratio columns).
(DOCX)

**S2 Fig. Arts/NI vs Prol/NI: Interaction network analysis of differentially downregulated genes.** Interaction network built using STRING knowledgebase (v. 12.0). Network was imported into Cytoscape (v. 3.10.2) and clustering analysis was performed with MCODE (v. 2.0.3) with degree cutoff 2, node density cutoff 0.1, K-core 2, and maximum depth 100. Resulting clusters were functionally annotated on STRING database.
(DOCX)

**S3 Fig. Ingenuity Pathway Analysis for IFN in NI neurons.** Upstream regulators network for IFNA2, IFNB1, and IFNG (A), and results from upstream regulators prediction and activation status. Intensity of red color in molecules indicates expression levels as shown in the legend. B) Table with prediction confidence and activation z-score value for each molecule.
(DOCX)

**S4 Fig. Gene set enrichment analysis (GSEA) plots.** Shows enrichment of molecular signatures from two group comparison between Arst/NI vs Prol/NI groups.
(DOCX)

**S5 Fig. Example of Wnt signaling in arrested CJ+ cells.** Overrepresentation analysis was done on g:Profiler web server and enriched pathways (FDR p < 0.05) were clustered using Enrichment map application on Cytoscape. Gene list to the left shows common upregulated genes across those enriched pathways.
(DOCX)

**S6 Fig. CJ+ cells: Ingenuity pathway analysis network.** Overlap between differentially expressed genes and anti-viral response induced by IFN. Activation is indicated by orange lines, and cellular location of the response genes are also shown. In CJ+ cells the IFNs are not extracellular in origin but made by the cell since no other cells are present to produce activating cDNAs. From RT/qPCR direct experimental results in CJ+ cells IFNs were far higher than in uninfected cells indicating above transcripts were enhanced and due to recrudescence of the infectious agent.
(DOCX)

**S7 Fig. Ingenuity pathway analysis of three different comparisons based on pathway activation status (z-score).** Note the difference for the CJ+/CJ− comparison lane 3 with Arst/NI in lane 1 and CJ+/NI showing underlying differences in CJ− brought out by arrest. Blue color indicates a negative z-score (inhibition), while orange color indicates a positive z-score and activation.
(DOCX)

**S8 Fig. Enrichment plots from Gene Set Enrichment Analysis (GSEA).** Shows enriched molecular signatures comparing CJ+ versus Prol/NI samples. Compare S4 above for gene enrichment analyses of Arst/NI versus Prol/NI.
(DOCX)

**S9 Fig. Venn diagram of unique 196 upregulated CJ+ unique genes (A) and 146 downregulated genes.** B shows 146 CJ+ unique down regulated genes. Please refer to the supplement S10 for complete list.
(DOCX)

**S10 Table. Complete lists of unique up and down regulated genes in CJ+ versus Prol/CJ− (pink), Prol/NI (light green) and Arst/NI (dark green).** Note the minimal differences in Prol/CJ− versus Prol/NI comparisons and abundance of unique differences by arrest as seen in Arst/CJ+ and Arst/NI (dark green columns).
(XLSX)

## Acknowledgments

We are indebted to Brett Robb and NEB for making the ribosomal-deleted RNA-seq libraries. We also thank Najoua Bolakhrif for making the Graphical overview and textual improvements, and Stephen-Francis Marino for critical suggestions on the manuscript.

## Author contributions

**Conceptualization:** Nathan Pagano, Laura Manuelidis.

**Data curation:** Nathan Pagano, Rolando Garcia-Milian, Laura Manuelidis.

**Formal analysis:** Nathan Pagano, Gerard Aguilar Perez, Rolando Garcia-Milian, Laura Manuelidis.

**Investigation:** Nathan Pagano, Gerard Aguilar Perez, Laura Manuelidis.

**Methodology:** Nathan Pagano, Gerard Aguilar Perez, Rolando Garcia-Milian, Laura Manuelidis.

**Project administration:** Laura Manuelidis.

**Software:** Rolando Garcia-Milian.

**Supervision:** Laura Manuelidis.

**Visualization:** Rolando Garcia-Milian.

**Writing – original draft:** Rolando Garcia-Milian, Laura Manuelidis.

**Writing – review & editing:** Nathan Pagano, Rolando Garcia-Milian, Laura Manuelidis.

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
