## [Decision Letter · Decision Letter 0]

28 Feb 2025

PONE-D-24-53655Proliferative arrest induces neuronal differentiation and innate immune responses in normal and Creutzfeldt-Jakob Disease agent infected rat septal neuronsPLOS ONE

Dear Dr. Manuelidis,

Thank you for submitting your manuscript to PLOS ONE. After careful consideration, we feel that it has merit but does not fully meet PLOS ONE’s publication criteria as it currently stands. Therefore, we invite you to submit a revised version of the manuscript that addresses the points raised during the review process.

The reviews for the above manuscript have been received. The reviewers' comments are included along with this letter.

Two independent reviewers have assessed the manuscript's suitability for publication and solicit additional clarification on certain outstanding issues.

Based on the reports of the reviewers and my own assessment, I recommend that the authors revise the manuscript to address the concerns raised.

We look forward to receiving your revised manuscript.

Kind regards,

Abhinava Kumar Mishra, PhD

Academic Editor

PLOS ONE

Journal Requirements:

Reviewers' comments:

Reviewer's Responses to Questions

**Comments to the Author**

1. Is the manuscript technically sound, and do the data support the conclusions?

Reviewer #1: Yes

Reviewer #2: Yes

2. Has the statistical analysis been performed appropriately and rigorously? 

Reviewer #1: Yes

Reviewer #2: Yes

3. Have the authors made all data underlying the findings in their manuscript fully available?

Reviewer #1: Yes

Reviewer #2: Yes

4. Is the manuscript presented in an intelligible fashion and written in standard English?

Reviewer #1: Yes

Reviewer #2: Yes

5. Review Comments to the Author

Reviewer #1: General Comments

This study investigates the impact of proliferative arrest on neuronal differentiation and innate immune responses in rat septal neurons, comparing normal cells to those infected with the Creutzfeldt-Jakob Disease (CJD) agent. It employs transcriptomic analysis to uncover changes associated with latent and productive CJD infection states. The findings highlight that proliferative arrest in normal neurons leads to upregulation of neuronal differentiation markers, anti-proliferative genes, and innate immune pathway genes.

In contrast, neurons with latent (CJ-) infection exhibit epigenetic imprinting without significant transcriptional differences from normal neurons. However, productive (CJ+) arrested neurons show suppressed neuronal differentiation, heightened interferon and cytokine responses, partial escape from proliferative control, and persistent high infectivity. The study proposes that latent CJD infections can cause lasting transcriptional and epigenetic changes, contributing to the silent persistence of infectious agents in peripheral tissues.

While the study is intriguing and relevant, it requires additional validation to strengthen its conclusions and enhance its suitability for publication.

Major Comments

1. Lack of Functional Validation for Transcriptomic Findings

The manuscript provides valuable transcriptomic data but lacks functional or biochemical validation of key findings. While reproducibility across biological replicates is mentioned, RNA-seq results need to be validated using methods such as RT-qPCR or Western blot. RNA-level changes do not always correlate with protein-level changes or cellular phenotypes, making functional validation essential. To strengthen the robustness of the bioinformatics conclusions, the authors should validate the expression levels of selected genes with significant fold changes using RT-qPCR or similar methods.

2. Restricted Focus on a Single CJD Strain and In Vitro Model

The study's focus on a single CJD strain (FU-CJD) limits the generalizability of its findings. Including additional CJD strains in the analysis would provide broader insights and enhance the study's applicability. Similarly, reliance solely on an in vitro cell culture model may not fully recapitulate in vivo conditions. Expanding the scope of the study to include other models or strains is recommended.

Minor Comments

1. Abstract Revision

The abstract should be concise and easily understandable. The current abstract is overly complex and includes unnecessary details, such as specific candidate gene names, which are better suited for the main text. Simplifying the abstract will make it more accessible to readers.

2. Discussion Length

The discussion section is excessively long (six pages), with repetitive elaboration on results. A discussion should provide a concise summary of the findings, their implications, and their alignment with existing literature. The authors should restructure this section to avoid redundancy and improve clarity.

Reviewer #2: This manuscript by Pagano et al. investigated how RNA expression varies in rat post-mitotic septal neuronal cell lines at arrested versus proliferative states in both infected and uninfected conditions. The study identifies several pathways that are associated with specific states and conditions which can provide the basis for future studies into the mechanisms of neural differentiation as well immune responses in neurodegenerative diseases. The study was rigorously performed and the findings are well presented.

I suggest some minor modifications:

1. The authors need to always provide the full form of terms while introducing them for the first time

2. Th authors need to provide a protocol for qPCRs in the materials and methods section and provide a list of all the primers used along with their sequences

3. The figures have been written as figure.tif at multiple occasions. The authors need to correct that. Also, the authors often tended to start sentences by 'The figure shows' which is a rather uncommon practice. Its better to just state the findings and mention the relevant figure in bracket.

4. The introduction requires some rewriting to improve clarity about the necessity of this endeavor. It might be better if the authors start with the disease aspect of this study and make a logical progression to justify the usage of SEP cells for this study. Then they can provide information about relevant studies in these cells and then proceed to the next section.

6. PLOS authors have the option to publish the peer review history of their article (what does this mean? ). If published, this will include your full peer review and any attached files.

**Do you want your identity to be public for this peer review?** For information about this choice, including consent withdrawal, please see our Privacy Policy .

Reviewer #1: **Yes: ** Avinash Chandel

Reviewer #2: No

---

## [Author Response · Author response to Decision Letter 1]

11 Mar 2025

We have made changes in the manuscript that address and clarify every point made by reviewers as detailed below. In addition, we made substantial changes in the introduction, discussion and conclusions to simplify the flow and logic of the findings. While redundancies were cut, added explanatory changes further highlighted 1) relevant corroborating animal experiments showing early brain upregulation of cytokine/chemokines and TGF-��transcripts in response to CJ brain infection (p21, first paragraph), 2) IFN linked secondary transcripts in CJ+ microglia and 3) extensive prior biological reversible DNA replication, neurodifferentiation, and Western blot and RT/qPCR assays in our publications that predicted most of the secondary global changes found here. We thank the reviewers for their queries and comments that helped improve the manuscript. All required formatting changes, including supplemental and figure captions have also been made. Please see complete response letter for details

---

## [Decision Letter · Decision Letter 1]

16 Apr 2025

Proliferative arrest induces neuronal differentiation and innate immune responses in normal and Creutzfeldt-Jakob Disease agent (CJ) infected rat septal neurons

PONE-D-24-53655R1

Dear Dr. Manuelidis,

We’re pleased to inform you that your manuscript has been judged scientifically suitable for publication and will be formally accepted for publication once it meets all outstanding technical requirements.

Kind regards,

Abhinava Kumar Mishra, PhD

Academic Editor

PLOS ONE

Additional Editor Comments (optional):

Reviewers' comments:

Reviewer's Responses to Questions

**Comments to the Author**

1. If the authors have adequately addressed your comments raised in a previous round of review and you feel that this manuscript is now acceptable for publication, you may indicate that here to bypass the “Comments to the Author” section, enter your conflict of interest statement in the “Confidential to Editor” section, and submit your "Accept" recommendation.

Reviewer #1: All comments have been addressed

Reviewer #2: All comments have been addressed

2. Is the manuscript technically sound, and do the data support the conclusions?

Reviewer #1: Yes

Reviewer #2: Yes

3. Has the statistical analysis been performed appropriately and rigorously? 

Reviewer #1: Yes

Reviewer #2: Yes

4. Have the authors made all data underlying the findings in their manuscript fully available?

Reviewer #1: Yes

Reviewer #2: Yes

5. Is the manuscript presented in an intelligible fashion and written in standard English?

Reviewer #1: Yes

Reviewer #2: Yes

6. Review Comments to the Author

Reviewer #1: The authors have effectively addressed the concerns raised and have also cited relevant previous studies to support their responses to some of the issues highlighted. The current version of the abstract and discussion has been significantly improved Thus, I find the current version of the manuscript to be well-prepared and suitable for publication.

Reviewer #2: The authors have satisfactorily addressed all my comments. I am satisfied with the current version of the manuscript and recommend it being accepted without further changes.

7. PLOS authors have the option to publish the peer review history of their article (what does this mean? ). If published, this will include your full peer review and any attached files.

**Do you want your identity to be public for this peer review?** For information about this choice, including consent withdrawal, please see our Privacy Policy .

Reviewer #1: No

Reviewer #2: No

---

## [Editor Report · Acceptance letter]

PONE-D-24-53655R1

PLOS ONE

Dear Dr. Manuelidis,

I'm pleased to inform you that your manuscript has been deemed suitable for publication in PLOS ONE. Congratulations! Your manuscript is now being handed over to our production team.

Kind regards,

on behalf of

Dr. Abhinava Kumar Mishra

Academic Editor

PLOS ONE